# Quantitative analysis of plant ER architecture and dynamics

Charlotte Pain [1], Verena Kriechbaumer [1], Maike Kittelmann [1], Chris Hawes [1] & Mark Fricker [2]

The endoplasmic reticulum (ER) is a highly dynamic polygonal membrane network composed of interconnected tubules and sheets (cisternae) that forms the first compartment in the secretory pathway involved in protein translocation, folding, glycosylation, quality control, lipid synthesis, calcium signalling, and metabolon formation. Despite its central role in this plethora of biosynthetic, metabolic and physiological processes, there is little quantitative information on ER structure, morphology or dynamics. Here we describe a software package (AnalyzER) to automatically extract ER tubules and cisternae from multi-dimensional fluorescence images of plant ER. The structure, topology, protein-localisation patterns, and dynamics are automatically quantified using spatial, intensity and graph-theoretic metrics. We validate the method against manually-traced ground-truth networks, and calibrate the sub-resolution width estimates against ER profiles identified in serial block-face SEM images. We apply the approach to quantify the effects on ER morphology of drug treatments, abiotic stress and over-expression of ER tubule-shaping and cisternal-modifying proteins.

[1] Department of Biological and Medical Sciences, Oxford Brookes University, Oxford OX3 0BP, UK. [2] Department of Plant Sciences, University of Oxford, South Parks Road, Oxford OX1 3RB, UK. Correspondence and requests for materials should be addressed to M.F. (email: mark.fricker@plants.ox.ac.uk)

Much of our understanding of the structure, function and dynamics of the endoplasmic reticulum (ER) in plants has come from live-cell imaging of the ER in epidermal cells using video-enhanced microscopy or after labelling with fluorescent dyes[1–3], or expression of transgenic reporters, such as GFP-HDEL[4–7]. Typically labelled ER is visualised in paradermal optical sections using confocal microscopy where the ER is constrained to an almost planar structure by the thin layer of cortical cytoplasm underlying the periclinal cell wall[5]. In general, the cortical ER forms a complex dynamic polygonal network of membrane-bounded tubules and flattened sheet-like cisternae that ramify throughout the cytoplasm[8–10]. However, the morphology of the tubules, the size and shape of the cisternae, and the proportion of tubules to cisternae varies during development[6,10–12], and under different experimental treatments[7,13]. For example, over-expression of members of the reticulon ER-shaping protein family[14–18], causes constrictions along the length of the tubules, and can convert cisternae to tubules[16]. Likewise, the morphology of the cisternae is affected by proteins, such as Lunapark 1 and 2, that can induce cisterna formation[19], and drug treatments, such as Brefeldin A (BFA)[13], which can vary the size and abundance of cisternae. The cisternae may not be simple uniform compartments. Thus, early in plant development the cisternae form large sheets punctuated by fenestrations[6,20], while in animal cells the distribution of ER lumen and tubule markers across cisternae fluctuates during live cell imaging, suggesting that cisternae may result from local appression of multiple tubules, resulting in a sub-resolution tubular matrix with internal spaces that can only be resolved with super-resolution techniques[21].

Quantitative measurements are further complicated in plants, as the ER network is highly dynamic, with rapid bulk streaming along actin cables, localised tubule extension, shrinkage, sliding and breakage, and expansion, contraction, reshaping and movement of cisternae[3,8,10,22]. While the velocity of discrete structures, such as the plant Golgi, can be readily quantified by tracking algorithms[22,23], alternative approaches based on optical flow[24], are required to estimate velocities for the ER. For example, local cross-correlation of small image patches was used to estimate ER movement between frames[25], and reported the maximum velocity around 1–2 μm s$^{-1}$. The maximum speeds in the thicker cytoplasmic strands running deeper into the cell range from 4 to 10 μm s$^{-1}$ [3,26], similar to the maximum speeds for other organelles[22]. Motility of the ER is also dependent on cell type and developmental stage, and typically increases during cell expansion[20].

In addition to the highly dynamic movement, the plant ER network also has a subset of static elements, such as ER-plasma membrane contact sites (EPCS), that act as fixed anchors to stabilise the polygonal network[27]. These features are visible at the electron microscopy (EM) level[28] and have been characterised with nanometre resolution using TEM and electron tomography[12]. They have been identified in living cells using persistency analysis[5,29–34], and following manipulation using optical tweezers[22,29,35]. Increasing the expression of certain proteins, such as VAP27 and SYT1, modulate the number and size of EPCS, and also affect the stability of the ER network[27,33,34].

Overall it is clear that, a complete description of the ER should include metrics quantifying morphology, persistency and dynamics. However, previous quantitative morphological measurements of relative volumes of cisternae and tubules in plants have been made in fixed tissue using stereological analyses of EM images[36,37]. More recently, EM tomography and 3-D reconstruction have been used to analyse ER structure in yeast cells[38], and in cultured mammalian cells using fluorescence and serial block face SEM[39]. However, measurements of dynamics require living tissues, and live cell imaging techniques. This presents a further challenge as the width of the ER tubules is expected to be 30–50 nm[9], which is below the resolution of standard confocal microscopes. Super-resolution techniques, such as stimulated emission depletion microscopy (STED), have been used to resolve structures on this scale, but are not yet routinely available nor compatible with dynamic imaging of ER movement[40]. Nevertheless, the width of the ER can be estimated from the labelling intensity, assuming the relevant marker is evenly distributed within the lumen, there is a linear relationship between fluorescence intensity and volume, and the tubule is fully contained within the point-spread-function (psf) of the microscope[41,42].

In addition to metric-based morphological measurements, topological measures of the ER network structure have been made from a graph representation of the network, where nodes represent junctions or free ends, and edges represent the interconnecting tubules[14,30,32,42,43]. Unlike morphological measurements, graph–theoretic measures reflect the connectivity of the ER, rather than the physical size of the components. They are therefore insensitive to errors in tubule width, but are critically dependent on extraction of a correctly connected network[43].

Given the wealth of information needed to characterise the structure and dynamics of the ER, and how these change with experimental treatments, we sought to develop an integrated analysis platform that combined existing measurement approaches into a coherent framework to provide complete quantitative information suitable for statistical evaluation. It is also important to enable rapid, medium throughput analysis to handle sufficient numbers of images to describe the variability present even within wild-type ER measurements. Here, we use a Zeiss 880 AiryScan confocal microscope to provide semi-high-resolution imaging of dynamic ER structures. The AnalyzER program then automatically quantifies (i) the length, width, morphology and protein distribution along ER tubules; (ii) the degree and branch angles at junctions (nodes) in the tubular network; (iii) the size, shape, and protein distribution in cisternae and around the perimeter of the cisternae; (iv) the topological organisation of the tubular and cisternal network determined using graph–theoretic metrics; (v) the distribution of immobile nodes, tubules and cisternae using persistency mapping; (vi) the local speed, direction, coherence, divergence and curl of movement of tubules and cisternae using optical flow; and (vii) the size and shape of the polygonal regions enclosed by the network.

The AnalyzER package is implemented in MatLab® (www.Mathworks.com) and available as a MatLab® app, or as a standalone package. It incorporates our previous tubule morphology analysis program[42], that was used to quantify tubule to analyse constrictions in the ER network caused by reticulon over-expression[14], but extends the analysis to quantitation of all aspects of ER structure and dynamics for multi-channel 4D time-series. The software package, manual, tutorial and test data sets are available from the Oxford Research Archive (ORA) (https://ora.ox.ac.uk/objects/uuid:cb0e2845-2a9c-495a-84f0-4dd2c5164463).

## Results

**Segmentation of tubules and cisternae.** All aspects of the analyses are handled through a single graphical user interface (GUI) to provide an integrated platform (Supplementary Fig. 1). The analysis approach is illustrated for transient expression of the ER lumenal-marker GFP-HDEL in leaf epidermal cells of *Nicotiana tabacum* (Fig. 1a). The ER is typically segmented from confocal optical sections using an intensity-based threshold to give a binary image, often using Otsu's method[43–45]. However, a global

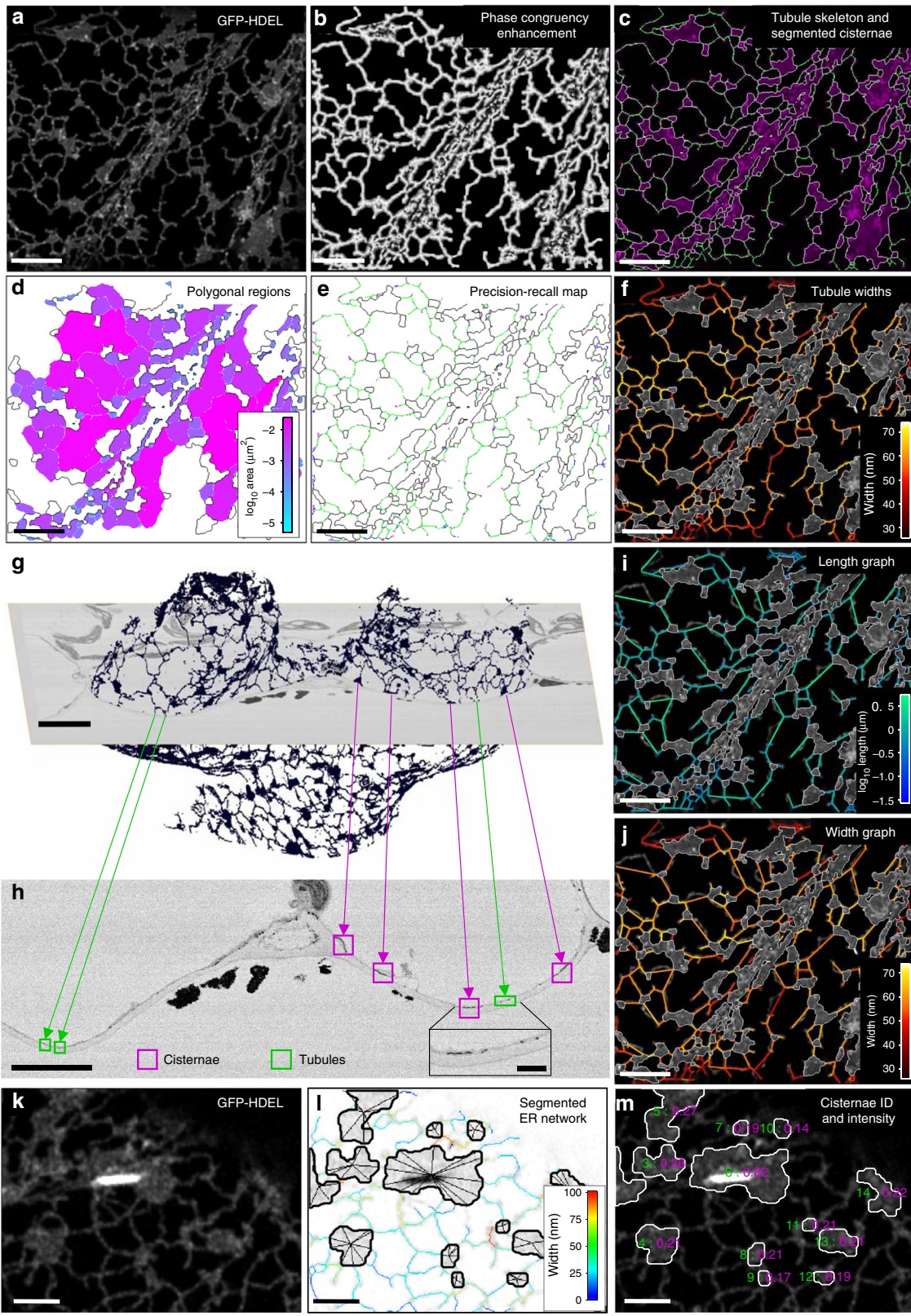

threshold excludes dimmer structures if it is set too high, or expands and fuses adjacent bright regions if it is set too low[43]. This is particularly problematic when the width, and hence intensity, of the tubules varies, for example, when analysing the effects of ER shaping proteins[14,42]. Segmentation of other biological network structures, including blood vessels[46], leaf veins[47], neurons[48], and fungal mycelia[49] use filters to highlight tubular

structures based on local image gradients. However, these approaches tend to fail at junctions and also retain a significant component of the original image intensity, making subsequent thresholding challenging. Thus, we have used intensity-independent phase-congruency analysis over a range of scales and orientations to provide additional tubule enhancement and subsequent robust segmentation[50–52].

**Fig. 1** Automated ER network analysis. **a** Analysis of a single tobacco leaf epidermal cell expressing the ER lumenal marker GFP-HDEL from a single paradermal optical section; **b** phase congruency enhancement of tubular elements using the local weighted mean phase angle; **c** hysteresis thresholding and thinning to produce a single pixel skeleton (green), and segmented cisternal structures (magenta with white boundary) following opening and active contour refinement; **d** polygonal regions pseudo-colour coded by $\log_{10}$ area; **e** comparison of the skeleton with a manually digitised ground truth (cisternae, black; true positives, green; false positives, blue; false negatives, magenta); **f** tubule width estimation using integrated granulometry; **g** A 3D reconstruction of the ER from an Arabidopsis leaf epidermal cell captured by SBF-SEM after zinc-iodine-osmium (ZIO) staining; **h** Enlargement of an SBF-SEM cross-section highlighting elements of tubular ER in green and cisternae in magenta, with a zoomed inset image (scale bar = 1 μm); **i, j** Graph representation of the ER network, with nodes at junctions linked by edges with a vector of properties including tubule length (**i**) and average width, excluding junctions (**j**). All scale bars = 5 μm, unless stated otherwise. **k** Arabidopsis plants stably expressing GFP-HDEL show high-intensity fluorescence concentrated in a single fusiform body within the ER; **l** segmentation of the cisterna containing the fusiform body using the lowest of multiple partitions of the image intensity histogram, with superimposed pixel skeleton colour-coded by width; **m** the maximum intensity value (magenta) is associated with a unique cisternal ID (green) which can be used to identify cisternae associated with fusiform bodies enabling separate quantification of their properties and behaviour. Scale bars = 2 μm

Tubule enhancement using the local weighted mean phase angle from phase-congruency analysis highlighted ridge-like elements, such as tubules, independent of their intensity (Fig. 1b), and ensured robust segmentation, even of tubules with low fluorescence, using subsequent hysteresis thresholding and thinning to give a single-pixel-wide skeleton (Fig. 1c). Sheet-like cisternae, regions of closely appressed tubules[21], or tubule/cisternal aggregates in moving streams were segmented independently using morphological opening (erosion followed by dilation) to remove the tubules[5]. The cisternal boundary was refined using active-contours to shrink the segmented region back onto the cisternal perimeter (Fig. 1c). The resulting objects were filtered to be above 0.3 μm², chosen to match the size of persistent puncta that are associated with immobile scaffold sites[5]. These puncta, although larger than the average junction in the network, were not considered as cisternae, but were included in the tubular network analysis as nodes. The other classes of non-tubular objects were not distinguished by morphology at this stage, but were classified using a combination of other features, such as speed and texture (see later). Completely enclosed polygonal regions within the ER network were also segmented from the complement of the combined tubule skeleton and cisternae (Fig. 1d).

We tested the accuracy of the automated segmentation against a set of manually-determined ground-truths using precision–recall analysis (Fig. 1e). Pixels matching the ground-truth were coded green, false positives coded blue and false negatives coded magenta, with a tolerance of half the minimum tubule diameter. In general, the automated extraction worked well for the central portion of any network (Dice Similarity Coefficient, $F_1 = 0.94 \pm 0.019$, mean ± s.e.m, $n = 5$), but there was more ambiguity at the margins where signals declined as the ER network moved out of focus following the curvature of the periclinal cell wall. To reduce these errors, we introduced additional masking around the boundary of the ER region using a contraction of the convex hull onto the irregular boundary, followed by erosion by 4–6 pixels to exclude low-level fluorescent structures on the periphery. The masked region was also used to define the area analysed for density-based measurements.

**Estimation of tubule width**. The initial estimate of the tubule width at each pixel in the skeleton was determined by granulometry techniques through a series of opening and closing operations with disk-shaped elements of increasing radii[51]. This builds up an intermediate $(x,y,s)$ image, where $s$ is the result of opening and closing at scale, $s$, which effectively maps the intensity of successive adjacent pixels back onto the skeleton in each layer in the $(x,y,s)$ image. The result is an intensity profile normal to the tubule in $s$ for each pixel in the skeleton. The width of each tubule was estimated from either the position of the

maximum (negative) gradient of the granulometry curve, or the integrated intensity under the curve as a measure of the relative amount of probe present locally (Fig. 1f). The actual width of tubules and cisternae were determined independently from ER profiles in transverse sections using serial-section block-face SEM (SBF-SEM) of Arabidopsis leaf tissue stained with zinc-iodide-osmium tetroxide (ZIO), and subsequent reconstructions with 90° rotation to give a planar view of the cortical ER (Fig. 1g, h). The average ER width from these measurements (40.51 ± 0.82 nm, mean ± SD, $n = 1$, 25 technical repeats) was used to calibrate the integrated granulometry intensity values using an idealised microscope psf[53], and the average fluorescence intensity from a cisternal sheet as an internal reference to standardise the intensity of fluorescence from a unit volume of ER.

**Conversion from a pixel-skeleton to a graph representation**. The single-pixel skeleton was converted to a graph representation with nodes at junctions, including puncta, and free ends, connected by edges (Fig. 1i, j). Each edge was associated with a vector of properties, including the length (Fig. 1i), and average width for the segment excluding the node regions (Fig. 1j), speed and persistency (see below). Likewise, each node was associated with a vector of properties including the node degree, branch angles between the incident tubules, speed and persistency (see below).

The cisternae were represented in the network graph as a single central node connected to each of the tubules incident on the boundary (e.g. Fig. 1l). A vector of properties was also associated with each cisterna, with selected values overlaid on the image. In the case of Arabidopsis and other members of the order Brassicales, segmentation and analysis of cisternae are complicated by the presence of fusiform bodies within the ER[54–57] that are strongly labelled by GFP-HDEL[56,58]. While the fusiform body can be segmented and quantified using automated image analysis[59], their presence has previously hindered segmentation of the surrounding ER network itself (Fig. 1k). Here, by partitioning the intensity histogram into multiple categories, typically 3 or 4, an appropriate threshold was selected automatically to extract the cisternae around the fusiform bodies (Fig. 1l). Cisternae containing fusiform bodies were easily identified later as their properties were significantly different from other cisternae. For example, in this image the maximum intensity for all cisternae was 0.21 ± 0.01 (mean ± SD, $n = 16$), compared to 0.82 for the cisterna with the fusiform body (Fig. 1m).

**Analysis of ER-shaping proteins and tubule morphology**. The default measurement of tubule width in the network graph returned an average along each tubule, excluding the overlap at

junctions (e.g. Fig. 1j). However, the intensity of luminal or membrane markers fluctuated along the length, indicative of changes in the (sub-resolution) width. Furthermore, the number, size and distribution of these bulges and constrictions altered during over-expression of ER-shaping proteins, such as reticulons[14–17]. To quantify these effects, the change in morphology along each tubules was analysed using the network graph to extract an intensity trace, integrated normal to the tubule axis, for both a GFP-HDEL luminal marker and an *Arabidopsis* RFP-tagged reticulon1 (Fig. 2a). The method used a peak-finding algorithm to identify the peaks along the intensity profile,

excluding the nodes, above a minimum height and sufficiently distinct from neighbouring peaks. The intervening troughs were defined similarly as peaks on an inverted intensity profile. These corresponded to bulges and constrictions, respectively, for the lumenal GFP-HDEL marker. Over-expression of RFP-RTN1 reduced the cisternal area and caused extensive constrictions along the ER tubules, with the lumenal GFP-HDEL marker restricted to discrete bulges. Analysis of the morphology for all tubules quantified the properties of the bulges and constrictions for GFP-HDEL and the degree of (negative) correlation with the distribution of the RFP-RTN1 (Fig. 2b). This can be seen clearly

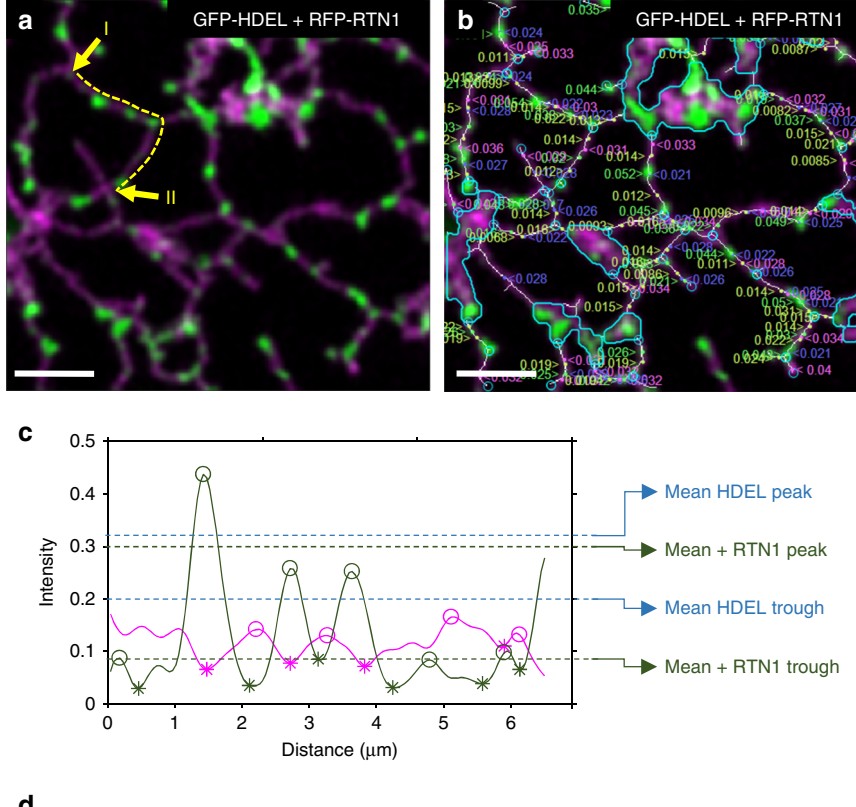

| | GFP-HDEL mean ± s.e.m. $n = 5$ | RFP-RTN1 mean ± s.e.m. $n = 6$ | GFP-HDEL | $t_{(9)}$ | $p$ value | |
|---|---|---|---|---|---|---|
| Peak density ($\mu m^{-1}$) | 1.085 ± 0.061 | 0.865 ± 0.041 | 0.990 ± 0.033 | −1.35 | 0.222 | |
| Trough density ($\mu m^{-1}$) | 1.868 ± 0.148 | 0.939 ± 0.068 | 1.402 ± 0.058 | −2.93 | 0.0309 | * |
| Peak intensity | 0.321 ± 0.024 | 0.184 ± 0.015 | 0.304 ± 0.019 | −0.55 | 0.600 | |
| Trough intensity | 0.200 ± 0.020 | 0.097 ± 0.011 | 0.089 ± 0.013 | −4.50 | 0.0027 | ** |
| Peak width ($\mu m$) | 0.0621 ± 0.0051 | 0.044 ± 0.0036 | 0.0544 ± 0.0048 | −1.45 | 0.188 | |
| Trough width ($\mu m$) | 0.0491 ± 0.0017 | 0.0334 ± 0.0031 | 0.0286 ± 0.0036 | −4.68 | 0.0014 | ** |
| Peak separation ($\mu m$) | 0.509 ± 0.029 | 0.628 ± 0.023 | 0.562 ± 0.021 | 1.49 | 0.175 | |
| Trough separation ($\mu m$) | 0.415 ± 0.018 | 0.624 ± 0.034 | 0.491 ± 0.0093 | 3.81 | 0.0084 | ** |
| Intensity ratio at peak | – | 4.33 ± 0.83 | 0.72 ± 0.15 | | | |
| Intensity ratio at trough | – | 1.95 ± 0.41 | 4.92 ± 1.03 | | | |

**Fig. 2** ER tubule morphology analysis. **a** A merged two-channel confocal optical section following transient over-expression of RFP-RTN1 (magenta) and GFP-HDEL (green) in tobacco leaf epidermal cells. **b** Automatic measurement of peaks and troughs for both RFP-RTN1 (left justified labels, peaks in magenta, troughs in blue) and GFP-HDEL (right justified labels, peaks in green, troughs in yellow). The underlying pixel skeleton is superimposed in white and the position of nodes indicated in cyan. Cisternae are also outlined in cyan. **c** Selected trace analysis between the yellow arrows in (**a**) showing the relative intensity for RFP-RTN1 (magenta) and GFP-HDEL (green), with the position of the peaks (open circles) and troughs (asterisks) indicated. The average values for peaks and troughs for the HDEL signal across all experiments are indicated in the presence and absence of RFP-RTN1. **d** Summary of tubule morphology metrics (mean ± s.e.m.) for GFP-HDEL alone ($n = 5$) and co-expressed with RFP-RTN1 ($n = 6$), with *T*-test statistic, associated *p*-value and significance, where *$p < 0.05$ and **$p < 0.01$. All scale bars = 5 µm. Source data are provided as a Source Data file

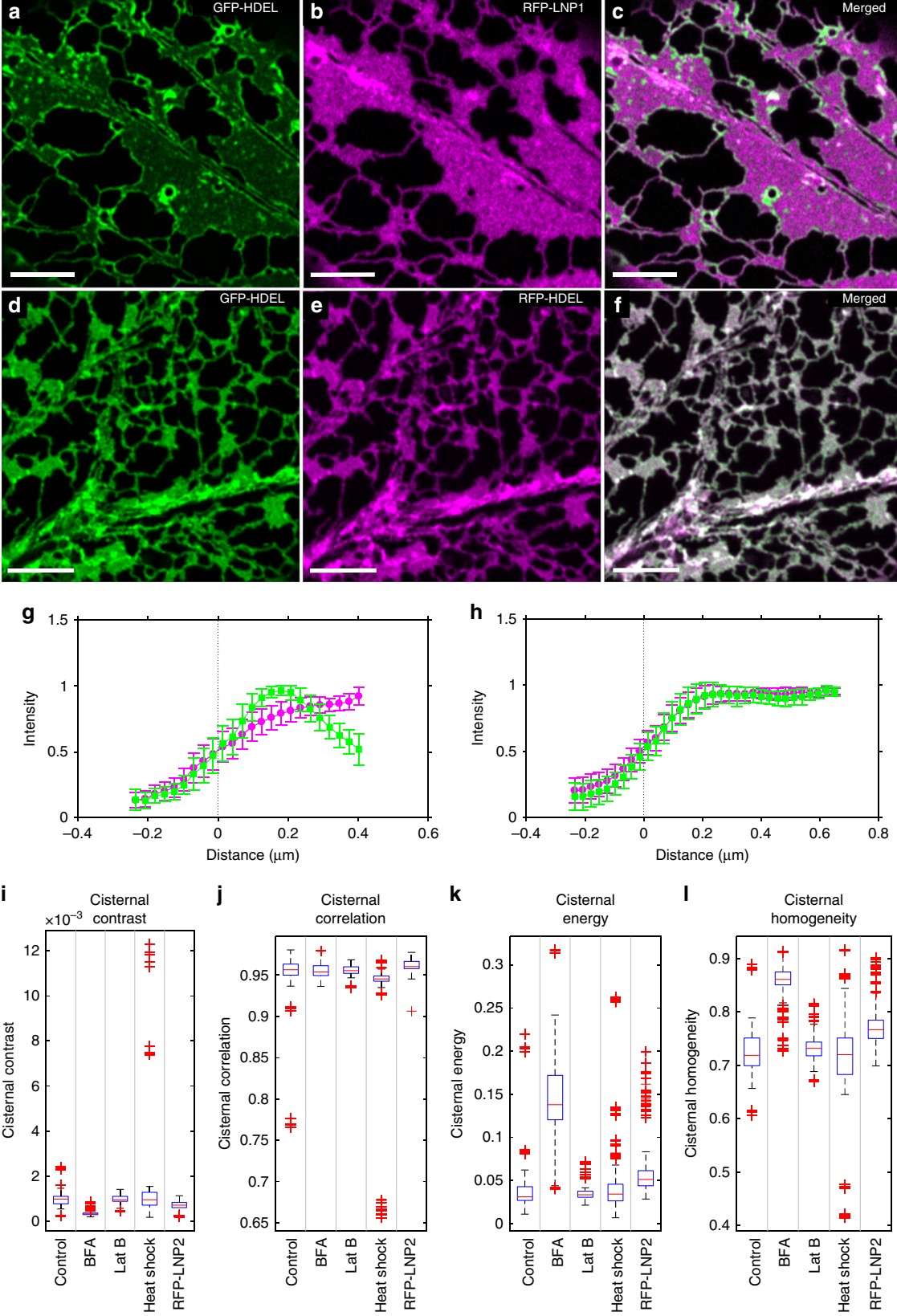

from the trace along a single tubule (Fig. 2c), between the yellow arrows shown in (Fig. 2a), where accumulation of RFP-RTN1 (magenta) caused a reduction in GFP-HDEL (green). In general, bulges in GFP-HDEL occurred every ~1 μm and peak intensity and width were not statistically different in the presence or absence of RFP-RTN1 (Fig. 2b, c). However, the troughs (constrictions) were significantly dimmer and thinner (Fig. 2d). Likewise, where two or more constrictions occurred on a single tubule, their separation was slightly, but significantly, increased in the presence of RFP-RTN1, leading to an overall reduction in

**Fig. 3** Analysis of label protein distribution in cisternae. Single optical sections showing transient co-expression of GFP-HDEL (**a**) and RFP-LNP1 (**b**), along with the merged image (**c**). The GFP-HDEL shows a peak in the distribution normal to the cisternal boundaries in the presence of RFP-LNP1 (**g**, mean ± SD, $n = 15$ cisternae). In contrast, co-expression of two lumenal markers GFP-HDEL (**d**) and RFP-HDEL (**e**) show complete co-localisation (**f**), and identical distributions across the boundary (**h**, mean ± SD, $n = 83$ cisternae). All scale bars = 5 µm. Comparison of ER cisternae texture metrics calculated from GLCM analysis: **i** contrast, **j** correlation, **k** energy, **l** homogeneity for GFP-HDEL and different treatments including BFA, Lat B, heat shock and transient overexpression of RFP-LNP2 (wild-type, $n = 46$; BFA, $n = 46$; Lat B, $n = 28$; heat shock, $n = 33$; RFP-LNP2, $n = 29$, where $n$ is the number of independent 5-frame time-series images). Box plot elements correspond to centre line, median; box limits, upper and lower quartiles; whiskers, 1.5× interquartile range; points, outliers. Source data are provided as a Source Data file

trough density. In the dual-labelled data, the ratio of intensities between the two channels at a bulge confirmed the anti-phase relationship between RFP-RTN1 localisation and GFP-HDEL distribution (Fig. 2c, d). Conversely, peaks in RFP-RTN1 localisation were associated with thinner tubules. The dual-labelling approach also confirms that the changes in GFP-HDEL intensity resulted from RFP-RTN1, not movement of the ER tubule out of the plane-of-focus.

**Mapping the localisation of proteins in cisternae.** Members of the reticulon family are also required along the periphery of the cisternae to induce curvature[9,38,60], while other proteins, such as *Arabidopsis* Lunapark 1 and 2, are suggested to maintain or induce the sheet-like structure[19]. Previous descriptions of protein localisation have been essentially qualitative, but these distributions can also be quantified. For example, following co-overexpression with RFP-LNP1, GFP-HDEL (Fig. 3a) was restricted to the perimeter of the cisternae (Fig. 3b, c), which resulted in a peak in the average intensity transect normal to the cisternal boundary (Fig. 3g). This was not due to non-specific crowding, as it was not observed with co-expression with RFP-HDEL as a second lumenal marker (Fig. 3d–f), where both markers showed identical profiles (Fig. 3h). Small flecks of GFP-HDEL were observed within the main sheet-like regions. These may arise from local bulges of the lumen within the sheet with a lower RFP-LNP1 density, or indicate accumulation along the cisternal perimeter around sub-resolution spaces[21], that may reflect the collapsed remnant of a previous polygonal region that has not yet undergone homotypic fusion[6,61].

**Texture-based analysis of cisternal sub-structure.** Quantifying sub-resolution spaces[21] or fenestrations[6] within cisternae or discriminating cisternae from a raft of appressed tubules, is challenging in living systems without recourse to super-resolution microscopy. Nevertheless, the different types of tubule and cisternal organisation predicted should leave a statistical imprint on the fluorescent intensities within the sheet-like regions, visible as a change in texture. Texture analysis has been widely used in medical imaging to identify features and patterns, often as inputs to machine-learning classifiers[62]. We therefore included a set of four standard orthogonal texture metrics (contrast, correlation, energy and homogeneity) calculated from a symmetric grey-level co-occurrence matrix (GLCM)[63,64], to describe the spatial distribution of intensities in each cisterna. The GLCM represents the pair-wise comparison of intensity values between each pixel and its neighbours at fixed geometric positions to give a joint probability distribution[63,64]. Thus, a completely homogeneous image would have a single entry on the diagonal of the GLCM. Noise causes non-zero entries to spread along and away from the diagonal, while texture increases the number of both on-axis and off-axis elements[63–65]. The structure of the GLCM can be summarised to give texture features that capture different aspects of the grey-level distribution. These features do not give a mechanistic understanding of sheet formation, but provide a set of

comparative measures between experimental treatments to identify which candidate ER-shaping proteins impact tubule-sheet transitions and/or cisternal organisation. Typical distribution patterns are shown in (Fig. 3i–l) for GFP-HDEL alone, following perturbation of the secretory pathway with Brefeldin A (BFA), physical stress through rapid heat shock (42 °C, 20 min), pharmacological disruption of the actin cytoskeleton with Latrunculin B (Lat B), or through transient over-expression of the ER shaping protein RFP-LNP2, which induces excessive cisternae. The contrast (Fig. 3i) is a measure of the variance within the sheet that gives a high weight to pixel pairs with dissimilar intensities. Correlation (Fig. 3j) is a measure of linear dependency of intensities and is bounded in the range [−1 1]. Energy, or angular second moment, (Fig. 3k) summarises the evenness of the spatial distribution, and is bounded in the range [0 1]. Homogeneity (Fig. 3l) is related to the inverse difference moment[64], and measures the closeness of the distribution to the diagonal. It is also bounded in the range [0 1]. Thus, a completely homogeneous cisterna has values of zero for contrast and energy, and one for correlation and homogeneity. In contrast, cisternae following BFA treatment show low contrast and high correlation and homogeneity, as expected for large homogeneous sheets, but the development of brighter patches within the cisternae is also observed as an increase in cisternal energy (Fig. 3k). This is also seen with RFP-LNP2, which increases the size and intensity variation within the cisternae.

**Persistency mapping of static elements.** The metrics described above can be calculated for a single optical-section to provide a snapshot of ER morphology, but analysis of ER dynamics in time-series images provides much more information. Persistency mapping characterises features that remain static, such as EPCS, and allows quantification of immobile tubules compared to mobile elements[5,8,45]. At present, there is no agreed definition of the duration an object has to remain in position to be regarded as persistent, but current practice has adopted a period of ~5–10 s for tubules and cisternae[5,45], or longer periods for persistent nodes[30,32]. For example, in time-series images (50 frames lasting 20.5 s) of GFP-HDEL transiently expressed in tobacco epidermal cells (Fig. 4a), a number of tubules remained stationary over the lag period examined (12 frames, ~5 s), with their persistence coded by intensity (Fig. 4d, g, green). Likewise, many cisternae were relatively stationary, with only fluctuations at the margin over this time period, unless they were translocating in the streaming cytoplasm (Fig. 4d, g, magenta). Nevertheless, certain nodes remained in position over a substantial fraction of the complete time series (in this case 10 s) in both the tubular network, and where tubules or cisternae moved over a fixed node (Fig. 4d, g, white crosses). Both Lat B (Fig. 4b, e, h) and heat shock (42 °C, 20 min) (Fig. 4c, f, i) increased the proportion of persistent nodes, tubules and cisternae, and the period that they remained static (Supplementary Movie 1a). In untreated cells, the majority of tubules (Fig. 5a) and nodes (Fig. 5d) showed low persistency, that increased in the presence of Lat B (Fig. 5b, e) or heat shock (Fig. 5c, f). Nevertheless, even with these

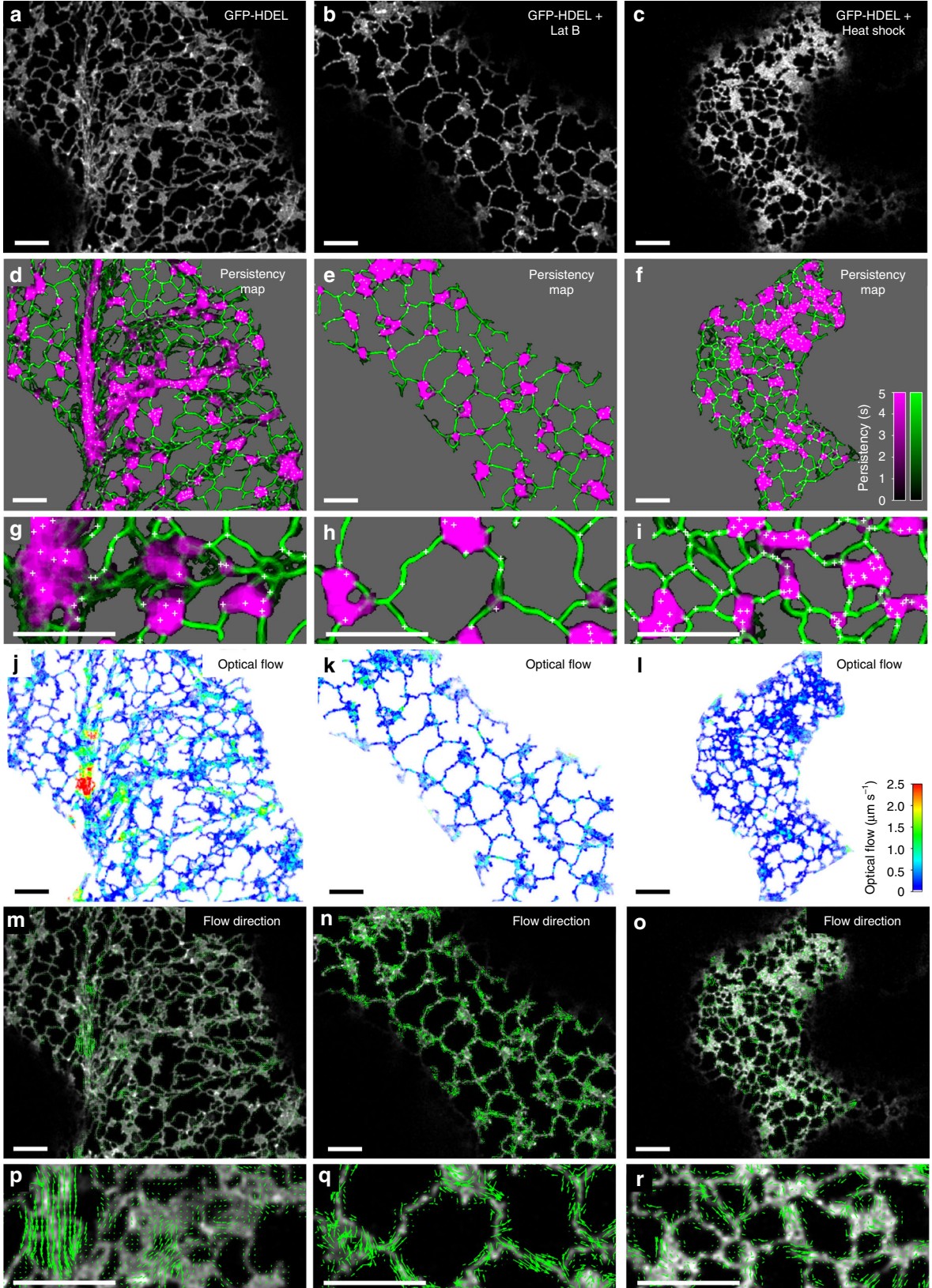

treatments a small number of tubules and nodes had low or zero persistency, indicative of substantial movement between frames. The cisternae in untreated cells were more stable with a peak in the distribution around 3.5 s (Fig. 5g, magenta), while Lat B and heat shock shifted most of the distribution to increased

persistency (Fig. 5h, i). Given the shapes of these distributions, the mean persistency alone does not fully capture the behaviour of the network, but division into categories of high and low persistence[5] are reasonable, and can now be made explicit using fits to the frequency distributions.

**Fig. 4** Mapping ER persistency and movement over time. The middle frame from a 50-frame time-series image taken at 0.41 s intervals of GFP-HDEL transiently expressed in tobacco leaf epidermal cells (**a**); after treatment with Lat B (**b**); or heat shock (42 °C, 20 min) (**c**). **d–f** persistency maps calculated using a 12 frame lag period (~5 s) for cisternae (magenta), tubules (green) and nodes (white crosses), with the background set to grey. The intensity corresponds to the length of time each pixel is occupied in the time series according to the inset scales; **g–i** Enlarged (3×) regions from the approximate centre of each image. **j–l** Optical flow analysis between the middle two frames of the same time series images for comparison. Images are pseudo-colour coded on the inset scale between zero (blue) and 2.5 μm s$^{-1}$ (red). **m–o** Predicted direction of ER network movement with arrows scaled to a maximum value of 2.5 μm s$^{-1}$. **p–r** Enlarged (3×) regions from the approximate centre of each image. All scale bars = 5 μm

**Quantifying ER velocity in time-series using optical flow**. The counterpoint to persistency measurements is characterisation of movement in the network. Previous work used optical flow measured by cross-correlation between local neighbourhoods[25], which provides an overall measure of movement, but does not distinguish between internal flows within the ER lumen or membrane, local ER remodelling, or bulk movement by cytoplasmic streaming[8,45]. Here we also characterise ER movement by optical flow, but using the Farnebäck algorithm, as this incorporates a scale-space analysis suited to quantify movement over different length scales, to give maps of speed (Fig. 4j–l) and direction (Fig. 4m–r), for untreated (Fig. 4j, m, p), Lat B (Fig. 4k, n, q), and heat shock (Fig. 4l, o, r) time-series (Supplementary Movie 1b, c). In untreated cells, there were small, localised movements from ER re-modelling and lateral wiggling of tubules through much of the network, with occasional substantial translocation of a bolus of ER along cytoplasmic strands (Fig. 4j). To quantify these speeds, the optical flow measurements were interrogated by the pixel skeleton and cisternal segmentation to extract the average local speed (scalar mean), average net speed (vector mean), and maximum speeds per tubule, node or cisterna. In all cases, the distribution of average speeds followed a log-normal distribution, with the back-transformed mean around 0.26 μm s$^{-1}$ for tubules (Fig. 5j), and 0.21 μm s$^{-1}$ for cisternae (Fig. 5l). The maximum speeds observed were higher, with means of 0.52 μm s$^{-1}$ for tubules (Fig. 5m), and 0.95 μm s$^{-1}$ for cisternae (Fig. 5l). Values greater than 2.5 μm s$^{-1}$ were routinely observed for ER moving along cortical actin strands (Fig. 4j, m, p). Movements of cisternae were also identified, although, as with other studies, it is not clear whether movement relates to translocation of the entire structure, or just elements within the structure[8]. Lat B (Fig. 5k, n) and heat shock (Fig. 5l, o) caused a reduction in the distribution of speeds, with the mean dropping by 24% (tubules) and 45% (cisternae), following Lat B treatment (Fig. 5k, n), or 34% (tubules) and 49% (cisternae) following heat shock (Fig. 5l, o). The tail of the distribution at higher speeds (>1 μm s$^{-1}$) also disappeared in these treatments and the direction of movement was less correlated, reverting to just local fluctuations, rather than directed flows (Fig. 4n, o, q, r). The extent of random movement to directed flow was quantified using flow coherence, calculated as the vector sum of velocities divided by the scalar sum of speeds (Fig. 5p–r). A value of 1 indicates all flows moving in parallel, compared to zero for no net movement. This confirmed that flows were directed in the HDEL control with high coherence (Fig. 5p), a shift to predominantly disorganised fluctuations in Lat B treatments (Fig. 5q), with heat shock intermediate (Fig. 5r). Local variation of flow within cisternal sheets and tubules was also quantified by changes in the curl (Supplementary Fig. 4d-f) and divergence (Supplementary Fig. 4g-i) of the flow pattern. Quantitatively, both curl and divergence of the vector flows showed greater spread in the HDEL control (Supplementary Fig. 5a, d) than in Lat B (Supplementary Fig. 5b,e) or heat shock (Supplementary Fig. 5c, f) treatments, but inspection of the corresponding images did not yield clear signature patterns that could be easily related to microscopic behaviour. We also note that different classes of cisternae can be

readily distinguished based on their morphology and dynamics that correspond to regions where the automated image analysis cannot distinguish sheets from aggregates or ribbons of tubules in the streaming cytoplasm (Supplementary Fig. 6a, b).

**Quantitative comparisons using multiple network parameters**. While examination of specific metrics such as tubule length, cisternal area, speed, coherence or persistency, can be used to test specific hypothesis about the role of a particular ER or cytoskeletal protein, calculation of a comprehensive set of metrics allows an un-biased comparison of any treatment with the GFP-HDEL control and provides potential input to automated classifiers to detect aberrant ER behaviour. This approach also captures subtle differences in behaviour that may not give a significant result for pairwise comparison of any single metric, but show a consistent phenotype as a combination of several metrics. The first stage of a parametric analysis explored here was based on multivariate analysis (MANOVA) to test the null hypothesis that the multivariate means were drawn from the same population, after transformation of individual metrics to achieve normality. As the null hypothesis was rejected, parametric ANOVA comparisons against the GFP-HDEL control were used to identify which metrics showed a significant difference. This helps to identify the potential site and mode of action of any given treatment as a guide to future investigation. The MANOVA also provided separation of the data by canonical discriminant variables to give a visual representation of the clustering relationships between the factors, and the potential for this approach for automated discriminant classifiers.

We illustrate this approach using a data set comprising 182, 5-frame movies of wild-type ER labelled with GFP-HDEL (Fig. 6a, $n = 46$), and with treatments including pharmacological perturbation with BFA (Fig. 6b, $n = 46$), and Lat B (Fig. 6c, $n = 28$), abiotic heat stress (Fig. 6d, $n = 33$), or transient over-expression of the ER shaping protein RFP-LNP2 (Fig. 6e, $n = 29$). Using a subset of 19 metrics, MANOVA analysis revealed the there was a highly significant difference in the structure and dynamics of the plant ER under these treatments (Pillai's trace, $F_{(72,648)} = 17.5$, $p = 2.7 \times 10^{-110}$; or Roy's largest root, $F_{(18,162)} = 36.7$, $p = 7.9 \times 10^{-48}$). The first two discriminant canonical variables from the MANOVA provided complete separation of Lat B and heat shock treatments from the other factors (Fig. 6f). The third canonical variable teased apart GFP-HDEL control from BFA (Fig. 6g), while the fourth canonical variable separated BFA, RFP-LNP2 and control (Fig. 6h). Subsequent ANOVAs revealed that BFA differed from control in almost every category of metric, apart from persistency and flow coherence. The main effects of Lat B and heat shock were on movement, as might be expected from disruption of the actin cytoskeleton, with some impact on the shape of cisternae and the intervening polygonal regions, but little difference in cisternal texture. Conversely, both BFA and RFP-LNP2 showed a significant difference from control in cisternal texture metrics of energy and homogeneity. Interestingly, while RFP-LNP2 predominantly affected shape and texture metrics, there was also a highly significant reduction in cisternal

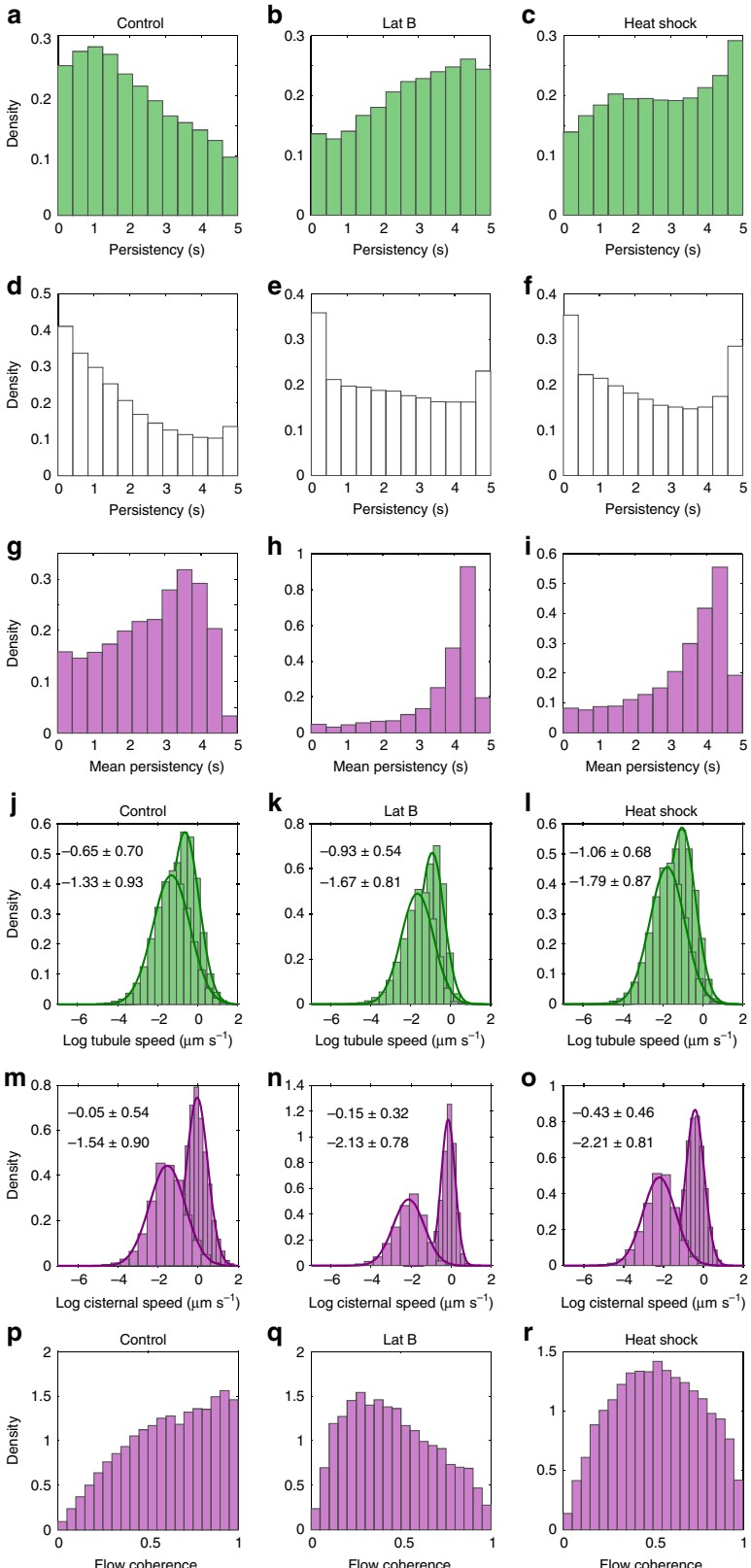

**Fig. 5** Distributions of persistency and optical flow analysis of ER dynamics. Probability density distributions of the mean persistency for tubules (**a**–**c**, green), nodes (**d**–**f**, white), and cisternae (**g**–**i**, magenta) for GFP-HDEL (**a**, **d**, **g**), and following treatment with Lat B (**b**, **e**, **h**) or heat shock (**c**, **f**, **i**). Probability density distributions of the log mean vector magnitude and log maximum speed of tubules (**j**–**l**, green) and cisternae (**m**–**o**, magenta), for the different treatments. Numerical values for the mean ± SD for the Gaussian fits to the mean (lower text) and max (upper text) distributions are inset. **p**–**r** Flow coherence for cisternae for the different treatments. (GFP-HDEL, $n = 6$, Lat B, $n = 8$, Heat shock, $n = 8$, where $n$ is the number independent 50-frame time-series images). Source data are provided as a Source Data file

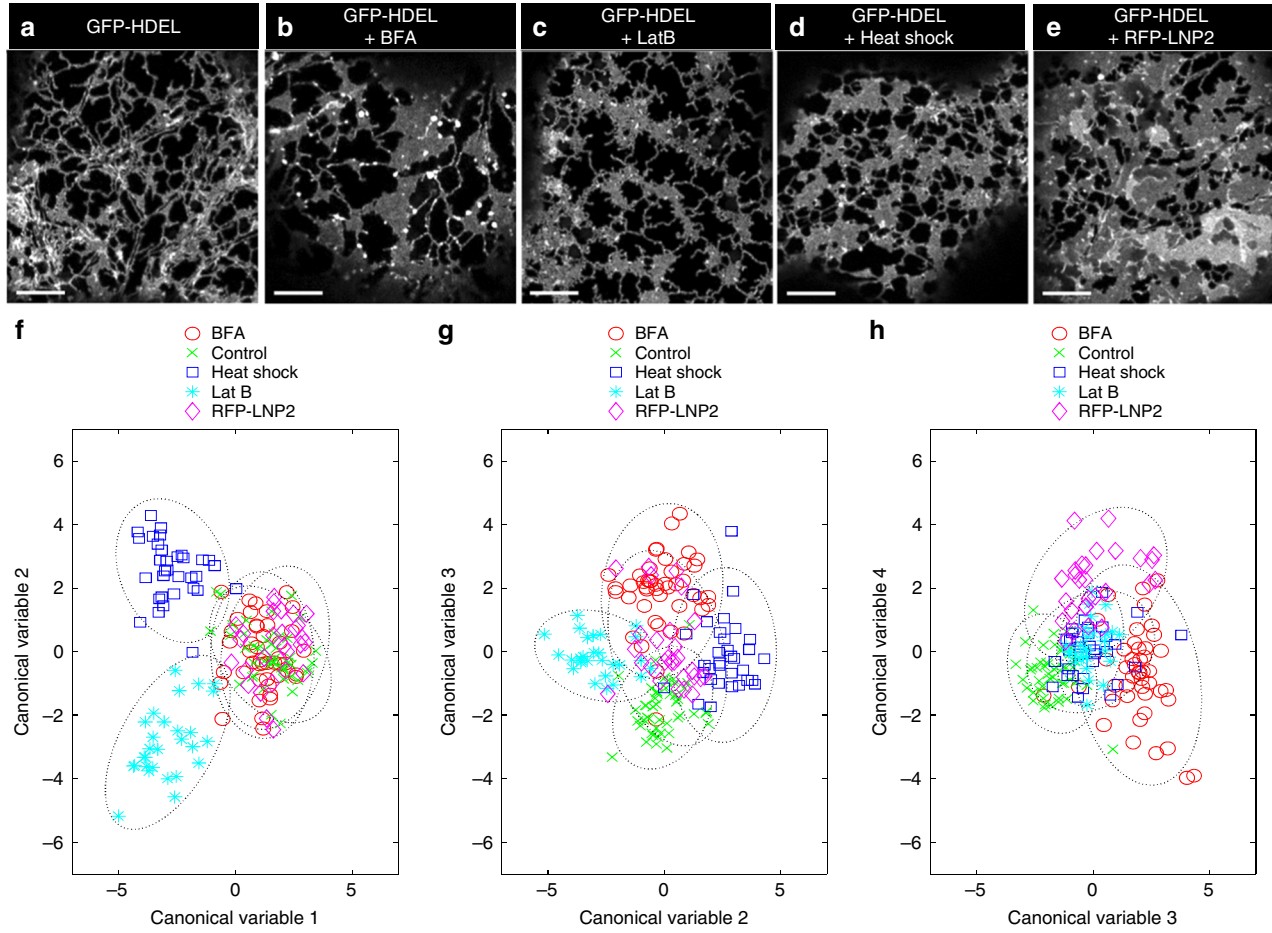

**Fig. 6** Multivariate analysis of ER morphology and dynamics. Representative single optical sections of tobacco leaf epidermal cells transiently expressing GFP-HDEL (**a**, $n = 46$); after treatment with BFA (**b**, $n = 46$); Lat B (**c**, $n = 28$); heat shock (**d**, $n = 33$); or over-expression of RFP-LNP2 (**e**, $n = 29$), where $n$ is the number of independent 5-frame time-series images. All scale bars = 5 μm. Pairwise scatter plots of the first 4 significant canonical variables from the MANOVA analysis grouped by treatment (**f**–**h**). The dotted line around each group represents the 95% confidence limit

speed, emphasising the importance of scoring across a wide range of both morphological and dynamic parameters when characterising ER-modifying protein function (Table 1).

## Discussion

We have developed a set of integrated tools that can be routinely used to assess a wide range of parameters describing the structure and dynamics of the ER network in living plant cells. We have validated the approach against manual ground-truth images, and tested performance by artificially inducing changes to the ER network in leaf epidermis through physical perturbation (heat shock), drug treatments affecting ER-Golgi trafficking (BFA), the actin cytoskeleton (Lat B), ER shaping proteins (RFP-RTN1), or cisterna-modifying proteins (RFP-LNP1 and RFP-LNP2). Using these diverse treatments as test cases, we can readily quantify the effects on ER-morphology and dynamics using a wide range of metrics that can both readily discriminate between the different treatments and provide a rich source of information to understand the physiological impact of each intervention. Importantly, we quantify the consequential effects that modification of one aspect of ER morphology, such as cisternal reorganisation, can have on overall network behaviour, such as ER dynamics. The software can also handle multiple channels, which allows object-based co-localisation of different reporter constructs, or quantitation of ratiometric physiological reporters, such as roGFP[66]. The analysis requires minimal user interaction, mainly to set

experiment-dependent parameters such as the background intensity, and can be run in batch mode to achieve a medium-throughput pipeline capable of handling hundreds of images required for statistical robustness or mutant screening. Taken together, this represents a major advance in un-biased, quantitative analysis of the plant ER, and should prove equally applicable to the analysis of mammalian or animal cortical ER.

## Methods

**Plant material and transient protein expression.** *N. tabacum* (SR1 cv Petit Havana) were grown in preparation for transient Agrobacterium-mediated expression as previously reported[67]. Transient expression of fluorescent constructs was performed according to Sparkes et al.[4]. In brief, transformed agrobacteria were pelleted by centrifugation at 2200$g$ at room temperature for 5 min. Infiltration buffer (5 mg ml$^{-1}$ glucose, 50 mM MES, 2 mM Na$_3$PO$_4$·12H$_2$O and 0.1 mM acetosyringone) was used to wash the pellet once and then to resuspend the *agrobacteria*. The bacterial suspension was diluted in the infiltration buffer to an OD$_{600}$ of 0.1 for GFP-HDEL, RFP-HDEL, RFP-LNP1 and RFP-LNP2 and an OD$_{600}$ of 0.05 for RFP-RTN1. The final dilution of the infiltration medium was injected through the stomata on the underside of the tobacco leaf using a 1 ml syringe. Infiltrated plants were kept at 23 °C for 3 days prior to imaging. For imaging of fusiform bodies, *Arabidopsis* Col-0 seeds stably expressing GFP-HDEL were sterilised in 70% ethanol and placed on 1/2 strength MS plates. These were stratified for 3 days at 4 °C in the dark and then grown in an incubator at 22 °C for 2 weeks prior to imaging.

**Cloning of expression plasmids.** Primers were obtained from Eurofins Genomics. Q5 high-fidelity DNA polymerase (New England Biolabs, Ipswich, MA, USA) was used for all PCR reactions. Using Gateway technology (Invitrogen, Waltham,

**Table 1 ANOVA for a subset of ER variables**

| ER structural subdomain | Variable | $F_{(4,177)}$ | p Value | BFA | Lat B | Heat shock | RFP-LNP2 |
|---|---|---|---|---|---|---|---|
| Tubule | Length | 15.78 | $4.58 \times 10^{-11}$ | *** | *** | *** | *** |
| Tubule | Maximum speed | 50.52 | $2.59 \times 10^{-28}$ | *** | *** | *** | *** |
| Tubule | Mean speed | 46.16 | $1.60 \times 10^{-26}$ | *** | *** | *** | *** |
| Tubule | Flow coherence | 19.71 | $1.96 \times 10^{-13}$ | | * | *** | |
| Tubule | Persistency | 60.36 | $5.11 \times 10^{-32}$ | | *** | *** | |
| Cisternae | Contrast | 5.894 | $1.78 \times 10^{-04}$ | * | | | |
| Cisternae | Correlation | 4.612 | $1.45 \times 10^{-03}$ | | | * | |
| Cisternae | Energy | 54.94 | $4.72 \times 10^{-30}$ | *** | | | *** |
| Cisternae | Homogeneity | 40.19 | $6.32 \times 10^{-24}$ | *** | | | *** |
| Cisternae | Maximum speed | 46.17 | $1.57 \times 10^{-26}$ | *** | *** | *** | *** |
| Cisternae | Mean speed | 74.1 | $8.72 \times 10^{-37}$ | *** | *** | *** | *** |
| Cisternae | Flow coherence | 77.42 | $7.61 \times 10^{-38}$ | *** | *** | *** | *** |
| Cisternae | Persistency | 53.46 | $2.00 \times 10^{-29}$ | ** | *** | *** | *** |
| Cisternae | Circularity | 8.325 | $3.56 \times 10^{-06}$ | ** | ** | ** | |
| Cisternae | Area | 6.467 | $7.03 \times 10^{-05}$ | ** | *** | ** | * |
| Cisternae | Perimeter | 15.56 | $6.29 \times 10^{-11}$ | *** | | | *** |
| Polygonal regions | Area | 40.14 | $6.61 \times 10^{-24}$ | *** | *** | | *** |
| Polygonal regions | Perimeter | 19.85 | $1.63 \times 10^{-13}$ | *** | *** | | *** |
| Polygonal regions | Circularity | 41.26 | $2.09 \times 10^{-24}$ | *** | *** | * | *** |

ANOVA results of planned comparisons against the GFP-HDEL control for the sub-set of variables included in the MANOVA, with the associated F-statistic and p-value. Means that were significantly different from the control using Tukey HSD post-hoc test are coded as *$p < 0.05$; **$p < 0.01$; and ***$p < 0.001$. (Wild-type, $n = 46$; BFA, $n = 46$; Lat B, $n = 28$; Heat shock, $n = 33$; RFP-LNP2, $n = 29$, where n is the number of independent 5-frame time-series images.) Source data are provided as a Source Data file.

MA, USA), genes of interest were cloned into the modified binary vector pB7WGR2,0[68] using the cauliflower mosaic virus 35S promoter upstream of coding fusions to RFP.

**Stress induction and drug treatments**. In preparation for stress and drug treatments, small leaf segments of approximately 25 mm² were cut from the transformed *N. tabacum*. Heat shock was induced by rapidly heating the leaf segment to 42 °C for 20 min. Drug treatments were performed by incubating leaf segments in either Lat B (25 µM for 30 min) or BFA (100 µg ml$^{-1}$ for 1 h) at room temperature.

**Confocal microscopy**. Leaf epidermal samples were imaged using a Zeiss PlanApo ×100/1.46 NA oil immersion objective on a Zeiss LSM880 confocal equipped with an Airyscan detector. Typically 512 × 512 images were collected in 8-bit with 2-line averaging at an $(x,y)$ pixel spacing of 20–80 nm with excitation at 488 nm (GFP) and 561 nm (RFP), and emission at 495–550 nm and 570–615 nm, respectively.

**Serial-section block-face SEM (SBFSEM)**. For EM, *Arabidopsis* Col-0 seeds expressing the Golgi marker ST-GFP were grown as described for 2 weeks. Seedlings were removed from the agar, cotyledon leaves cut off and submerged in fixative (1% paraformaldehyde, 1% glutaraldehyde, 2% sucrose and 2 mM CaCl$_2$ in 0.1 M sodium cacodylate buffer) under mild vacuum for 1 min and for a further hour without vacuum at room temperature. Leaves were then washed twice with 0.1 M sodium cacodylate buffer and once in deionized water for 10 min. Zinc-iodide was prepared by adding 3 g of zinc powder and 1 g of resublimed iodine to 20 ml distilled water with stirring for 10 min. The solution was filtered through filter paper and mixed 1:1 with 2% aqueous osmium tetroxide. Leaves were incubated in the ZIO mix overnight at room temperature with rocking, then washed with deionised water and dehydrated in an ethanol series from 10 to 100%. Infiltration with Spurr resin (hard) was carried out in a series from 10 to 100% for a minimum of 2 h for each step. Infiltration with 100% Spurr resin was exchanged 3 times over 2 days. Leaves were embedded in 100% Spurr resin in flat embedding dishes and polymerised at 70 °C for 11 h.

Leaves were mounted for cross-sectioning onto 3View stubs (Gatan, Abingdon, UK) with conductive epoxy (Chemtronics, Hoofddorp, Netherlands) and hardened for 4 h at 100 °C. The trimmed block was sputter coated with gold for 30 s to give a layer of around 20 nm to improve conductivity. Serial block face (SBF) images were collected with a Merlin Compact scanning electron microscope (Zeiss, Cambridge, UK) with the Gatan 3View system using a 30 µm aperture at 4 kV in variable pressure mode (50 Pa), a pixel dwell time 2 µs, and a pixel size of 6.9 nm. Serial sections were cut at 100 nm thickness.

SBF images were stacked, aligned, scaled to common mean and standard deviation and cropped to the area of interest using the IMOD software package[69]. Reconstruction of the ER was performed with Amira software (Thermo Fisher Scientific-FEI) using manual masking and thresholding tools. The rendering was visualised through surface generation and surface view in Amira.

**Image processing**. The GUI interface for ER image processing and analysis is shown in Supplementary Fig. 1. A complete description of all the processing steps and additional options available in the software is given in the accompanying manual available from the Oxford Research Archive (ORA) (https://ora.ox.ac.uk/objects/uuid:cb0e2845-2a9c-495a-84f0-4dd2c5164463). To standardise all subsequent processing and filtering operations, images were up-sampled using linear interpolation to ensure that the minimum tubule size, estimated from the full-width-half-maximum height (FWHM$_{min}$) of manual transects across the tubules, was around 5 pixels. Images were then background subtracted, using an average value determined from a manually-defined region-of-interest (ROI), and smoothed with a guided anisotropic filter[70], with the kernel size set to the FWHM$_{min}$ to avoid blurring across tubule edges. Regions where the signal decreased as the curvature of the cell took the ER out-of-focus, were masked where the pixel intensity fell below the lowest automatically determined threshold, after partitioning the image histogram into two, three or four sections using Otsu's method which minimises the intra-class variance within the pixel intensity histogram[44]. Multi-threshold partitioning was useful when the image included additional strongly labelled structures, such as Arabidopsis fusiform bodies[56,57,59]. Internal polygonal regions within the mask were initially filled to retain all internal structures, the outermost boundary identified by contraction of the convex hull using a shrink factor of 0.5–0.8, and the final boundary eroded by a fixed number of pixels, typically 4–6. The boundary was used to determine the area analysed for density-dependent metrics. The internal holes were then reintroduced to complete the mask for subsequent image processing operations

**Tubule enhancement and segmentation**. A number of options are included in the software to enhance the ER tubules including Vesselness[46], Neuriteness[48], second-order anisotropic Gaussian kernels (SOAGKs)[49], or intensity-independent phase-congruency filtering[50,71], typically applied over 3–5 spatial scales and 4–6 orientations. Although the phase-congruency filters characterise a number of relevant image properties, we have found the local weighted mean phase angle (Feature-Type) provided the most robust subsequent segmentation in a variety of contexts[51,52], including enhancement of ER-tubules[14,42]. Full details of the phase congruency analysis and alternative approaches to ridge enhancement are given in the accompanying manual available from ORA. The Feature-Type image was normalised to [0 1], and the tubular network segmented using hysteresis thresholding, with a lower threshold of 0.3 and upper threshold of 0.5. To ensure closely appressed regions were not merged during the hysteresis threshold, all local intensity minima were first identified using an h-minimum transform, with a threshold of 0.05, and set to zero. The binarised imaged was reduced to a single-pixel-wide skeleton by thinning using the Zhang and Suen algorithm[72].

**Segmentation of cisternae and polygonal regions**. ER cisternae were automatically segmented by image opening[5,8,45] to remove all features smaller than the maximum tubule diameter (estimated as the full-width half-maximum (FWHM$_{max}$) from manual transects across 3-way junctions). Segmentation was followed by an active-contour step[73] to shrink the segmented boundary back onto the intensity profile of the cisternae. The resulting objects were filtered on the basis of their area to be above a user-defined value of 0.3 µm², chosen as the area of

persistent puncta that are associated with immobile scaffold sites[5]. These puncta were included as nodes in the tubular network. A variety of morphological metrics (area, perimeter, major and minor axis length, solidity, elongation, circularity, and roughness) were determined for each sheet-like region (Supplementary Figure 2a). A similar set of morphological metrics was calculated for the polygonal regions enclosed by the ER network, following segmentation from the complement of the combined pixel skeleton and cisternae (Supplementary Figure 2b).

**Precision–recall analysis of segmentation performance**. Ground truth images were manually digitised from 5 single confocal images using a Wacom DTK-2241 digitising tablet (http://www.wacom.com/) and GIMP software (https://www.gimp.org/). The manual skeleton was thinned and masked in parallel to the automated skeleton, and the cisternae excluded. The number of true positives ($T_P$), true negatives ($T_N$), false positives ($F_P$) and false negatives ($F_N$) were calculated, allowing a tolerance of half $FWHM_{min}$[74]. Performance was assessed using precision–recall (P–R) analysis in preference to receiver operating characteristic (ROC) plots, as the former is better suited to imbalanced data sets, where the number of $T_N$ from the background is expected to be much greater than $T_P$ from the skeleton[75]. Precision ($P$) was calculated as $= \frac{T_P}{T_P + F_P}$, recall ($R$) as $R = \frac{T_P}{T_P + F_N}$ and overall performance assessed with the Dice Similarity Coefficient or $F_1$ score as the harmonic mean of precision and recall, $F_1 = 2 \times \frac{P \times R}{P + R}$.

**Estimation of tubule width**. The width of the tubules was estimated by several different methods. To estimate the tubule FWHM, the peak height was determined from the intensity, sampled for each pixel in the skeleton; the distance from the skeleton was estimated from the Euclidean distance transform (EDT) of the pixel skeleton; and the 50% threshold estimated from where the pixel intensity fell below half the peak intensity of the background-subtracted image. This gave the FWHM of the tubule convolved with the psf. Alternatively, a granulometry approach[51,52] was used whereby the intensity image was processed through a set of image opening operations (erosion followed by dilation), with a circular kernel of radius 0 to $FWHM_{max}$. This resulted in an intermediate $(x,y,s)$ image, where $s$ increased with the size of the disk-shaped kernel, that effectively builds a stack of the intensity of each successive neighbouring pixel underneath each pixel in the skeleton. The gradient profile for a given skeleton pixel in $s$ decreased slowly as the kernel sampled more of the object, but then reduced dramatically once the boundary of the object was reached, and the kernel sampled the background. The width was estimated as either the maximum (negative) gradient of this profile, or from the integrated intensity under the curve. However, in the former case, the radius was constrained to integer pixels values and results were limited by the discrete digital approximation of small kernels to a true disk-shaped kernel. Conversely, the latter provided an estimate of the local amount of fluorescence in the tubule which was related to the width, even if it was sub-resolution, provided it was assumed that the fluorescent probe was evenly distributed throughout the ER, there was a linear relationship between signal and the amount of probe, and the ER was fully within the sampling volume of the confocal defined by the psf[53]. The average intensity from the cisternal sheets ($I_s$), typically 0.35, the estimated sheet thickness ($T_s$) of 40 nm, and the size of the lateral psf ($psf_{xy}$) of 140 nm, were used to relate the measured intensity to the volume of ER that a fully-filled psf would yield when sampling sheets of defined thickness ($V_s = \pi psf_{xy}^2 T_s$) (Supplementary Figure 3). The radius of the tubules ($r_t$) was estimated by assuming the volume of the tubule sampled by the same stylised psf ($V_t = \pi r_t^2 psf_{xy}$) compared to the sheet, scaled as the ratio of the intensities from the tubule and sheet ($I_t/I_s$).

Thus, the radius of the tubule ($r_t$) was estimated as $r_t = \sqrt{(I_t/I_s)\left(psf_{xy}T_s/4\right)}$

(Supplementary Figure 3).

**Conversion to a graph representation**. The pixel skeleton was converted to a graph representation with nodes at the junctions and free ends, connected by edges along each tubule[30,42,43,51,52] (Supplementary Figure 2d(i)), using the *edgelink* algorithm written by Kovesi[71], which exports a list of pixel co-ordinates for each edge. Duplicate edges connecting the same two nodes were resolved to two separate edges in the graph by introducing an additional node with degree 2 in one arm (Supplementary Figure 2d(ii)). The pixel co-ordinates for each edge were used to extract relevant information from different images, such as intensity, width, speed, direction, coherence, divergence, curl or persistency, and the mean (or other summary statistic) and SD for each metric stored in a vector of properties for each edge. The initial width was further refined by excluding pixels within a certain distance from each node, to give the centre-weighted width, which provides a better estimate of the tubule width that excludes the 3-way junctions or cisternae (Supplementary Figure 2d(iii)). The exclusion distance was determined as the maximum of the initial widths of tubules connected to the node. A vector of properties was also associated with each node, including the degree, widths of the incident tubules, branch angles between incident tubules (estimated from segments drawn between the node and the mid-point of each connected tubule), strength (measured as the sum of the centre-width of the incident edges), persistency and speed (Supplementary Figure 2c). Summary metrics for orientation and branching angles were calculated using circular statistics[76]. The cisternae were represented as

a single-node placed at the intensity-weighted centroid of each cisterna, and connected to each edge incident on the boundary (Supplementary Figure 2d(i)). Edges within the cisternae were set to the average centre-width of all the tubules, along with their Euclidean length, for graph–theoretic calculations where the connectivity of the network is important, but were excluded from calculations of tubule statistics.

**Variation in morphology along the length of tubules and cisternal edges**. The position of bulges along the tubules (Supplementary Figure 2c) were estimated from peaks in the integrated intensity trace along each tubule, excluding the nodes, at each wavelength that were at least 5% of the maximum intensity (peak height) and also 3% of the maximum intensity greater than the signal on either side (peak prominence). Likewise, constrictions were determined using the same peak-finding process on the inverted intensity profile. The width of the tubule at these points was then extracted from the granulometry width image. The distribution of fluorescence around the perimeter of the cisternae was determined in the same manner.

**Characterisation of cisternal sub-structure using texture metrics**. The variation in intensities within the cisternal region was characterised using texture analysis from the normalised grey-level co-occurrence matrix (GLCM)[63–65]. In brief, for each pixel in the segmented cisternal region, the GLCM examined pixels at a distance that was set by the tubule radius to maximise the likelihood of revealing texture from appressed tubule regions. Results were aggregated in four directions (NW, N, NE, E) with symmetry to complete the angular sampling and ensure results were independent of orientation. Results were added to an accumulator array where the row ($i$) and column ($j$) indices correspond to the intensity value of the target pixel and the neighbour, grouped into a number of equal intensity bins to capture key differences in intensity without rendering the GLCM too sparse. The resultant GLCM was normalised to give a probability $p(i, j)$ of co-occurrence for each intensity pair. Choice of the number of intensity bins and spacing is known to affect texture features[65], thus to standardise results[65], images were collected under identical conditions, GLCM constructed using 32 fixed intensity bins over the full intensity range, at a distance equal to the minimum tubule radius, and with inclusion of all neighbouring pixels to give direction invariance. The contrast, $\sum_{i,j} |i - j|^2 p(i,j)$; correlation, $\sum_{i,j} \frac{(i - \mu i)(j - \mu j)p(i,j)}{\sigma_i \sigma_j}$, where $\mu i$ and $\mu j$ were the weighted mean intensities and $\sigma i$ and $\sigma j$ their standard deviation; energy, $\sum_{i,j} p(i,j)^2$; and homogeneity, $\sum_{i,j} \frac{p(i,j)}{1 + |i-j|}$, of the GLCM were calculated individually for each cisterna and then averaged, and also as a single accumulated GLCM for all cisternae, which effectively weights the contribution of each cisterna to the overall statistic by the number of pixels. Contrast measures the local variation in the GLCM, correlation measures the joint-probability distribution, energy measures the sum of the squares, while homogeneity measures how close elements in the GLCM are to the diagonal. Conveniently, correlation is scaled from $[-1\ 1]$, while energy and homogeneity are scaled between $[0\ 1]$. An idealised cisternal sheet would have a value of 1 for correlation and homogeneity, and zero for energy. The contrast scales from 0 to $(nbins - 1)^2$, but was normalised here by $(nbins - 1)^2$ as all images were processed with the same number of bins, to constrain the range to $[0\ 1]$, where an idealised sheet would have a value of zero.

The distribution of fluorescence intensities across the cisternal boundary was determined by averaging intensities in incrementing (external) and decrementing (internal) integer radial bins, determined from the EDT, normal to the edge of each cisternal region.

**Measurement of movement using optical flow**. A number of different methods are available in the software that draw from the MATLAB® Computer Vision Toolbox including Horn–Schunck[24], Lucas–Kanade[77] and Farnebäck[78]. Here, we have used the Farnebäck algorithm[78] which uses local quadratic polynomial expansions to approximate image intensities over a given neighbourhood, with weightings based on the strength of the signal and the distance from the central pixel. The algorithm also includes a coarse-to-fine pyramidal scale-space iteration to improve the a priori estimate of the initial displacement field, which allows the method to handle larger displacements. Typically a neighbourhood of 5 pixels was used, with a three-level image pyramid. The resultant vectors were then averaged over a 15-pixel region to estimate the local velocity. The average velocity for tubules and cisternae was calculated as the scalar mean to give an estimate of the total amount of movement irrespective of direction, while the net directed movement was estimated from the vector mean. The maximum speed was also recorded. Flow coherence was calculated as the vector mean divided by scalar mean, and gives a ratio of one for fully directed flow and zero for random movement. Divergence and curl were calculated from the velocity field. Summary metrics for optical flow direction were calculated using circular statistics[76].

**Persistency mapping of static elements**. In the original approach to persistency mapping[5], tubule and cisternal persistency were estimated from the difference between two images with a lag of 5–8 frames (8–14 s), following by a Boolean AND operation between time-points to only include features present in both. Results

were normalised to the total segmented area present in all frames (Boolean OR operation on the image stack). Here we implement two approaches. The first method followed Sparkes et al.[5] using the difference in the intensity image over a user-defined time-period, masked by a thresholded binary image from both time-points. The second used the segmented skeleton and cisternal regions directly. Binary images of the skeleton and cisterna were dilated by $FWHM_{min}/2$ and analysed separately or in combination, by either summing over a defined time window, giving a graded estimate of persistency, or by differencing between the start and end of the time window, giving a binary estimate of persistency. Results were displayed as a persistency map, summed over the entire time course, and were also aggregated for each edge and cisterna and included in their vector of properties. Nodes that were persistent over longer periods that might transiently be associated with tubules or cisternae moving across the anchor points[30,32], were identified following a 1D median filter in time across the entire time course followed by Gaussian smoothing in $xy$ with $\sigma = FWHM_{min}/2$, normalisation to [0 1], and extraction of the position of local maxima above an intensity threshold of 0.3–0.5.

**Statistical methods.** Results for different transgene expression and drug treatments were compared against a GFP-HDEL control using a parametric MANOVA with Pillai's trace and Roy's largest root as test statistics. Metrics that were bounded in the range [0 1] were arcsin transformed to improve normality, while those spanning a wide range, such as length, area or speed, were typically normalised using a log, logit or square root transform. Logit data was adjusted to the interval 0.025–0.975 before transformation. We note that some metrics, such as persistency, may be problematic as they deviate strongly from normality and cannot be reduced easily to a single summary statistic (mean, mode or median) that is well behaved. We also note that collectively, the co-variances may fail Mauchly's test for sphericity, which then requires additional correction factors to apply to the degrees of freedom before calculation of $p$-values. Nevertheless, the $F$-ratio and $p$-values we have encountered so far are all so low and highly significant (typically $< 10^{-30}$), that the conclusions are insensitive to these corrections. Parametric ANOVA comparison between individual means and the control used Tukey–Kramer HSD with results reported at least at the 95% significance level. Other statistics are given as mean ± SD when results are illustrative for a single experiment, or mean ± s.e.m. when averaged across experiments for a given $n$.

**Software implementation and code availability.** All of the analysis routines were implemented in MATLAB (The Mathworks) and are available in a standalone package for 64-bit Windows 10, or a MATLAB 2017a app from the Oxford Research Archive (https://ora.ox.ac.uk/objects/uuid:cb0e2845-2a9c-495a-84f0-4dd2c5164463). The app also provides access to the source code when installed in the ../MATLAB/Add-Ons/Apps/AnalyzER_v1_app/code folder.

**Reporting summary.** Further information on experimental design is available in the Nature Research Reporting Summary linked to this article.

## Data availability

All images, parameter files used in the software to analyse them, and the results are available from the Oxford Research Archive (ORA) (https://ora.ox.ac.uk/objects/uuid:cb0e2845-2a9c-495a-84f0-4dd2c5164463).

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

## Acknowledgements

This work was supported by the Human Frontier Science Program (RGP0053/2012, M.F.); the Leverhulme Trust (RPG-2015-437, M.F.); the Institute of Advanced Studies, Durham (M.F.); BBSRC (BB/M000168/1, C.H.) via the ERA-CAPs programme Per Aspera; BBSRC 17ALERT (BB/R014086/1, M.F.); The John Fell Oxford University Press (OUP) Fund (M.F.); a Vice Chancellor's Research Fellowship from Oxford Brookes University (UK) to V.K. and a BBSRC DTP studentship to C.P. (BB/M011224/1).

## Author contributions

V.K., C.H. and M.F. conceived the work, C.P. performed the confocal experiments and analyses, M.K. performed the SBFSEM, V.K. and C.H. supervised the experiments, M.F. wrote the software and performed the analyses, C.P. wrote the tutorial, C.P., V.K., C.H. and M.F. wrote the paper.

## Additional information

**Competing interests:** The authors declare no competing interests.

