## [Peer Review File · Nature Communications]

Reviewers' comments:

Reviewer #1 (Remarks to the Author):

This paper describes a Matlab-based, integrated set of software tools to analyze the changes in the [cortical] ER as it moves across the paradermal cytoplasm of mature plant epidermal cells and transitions between tubules to sheets. It incorporates interesting approaches to estimate the sub-resolution diameter of the ER and bases those estimations on measurements of serial block-face 3D reconstructions from ZIO impregnated tobacco leaf. Some improvements of the existing analysis techniques include a treatment of brightly-fluorescent ER bodies that move along with the ER tubules in Arabidopsis (but Arabidopsis is not included in the plant material section) and implementation of a better optical flow algorithm (Farneback) for looking at movement of and within tubules. Persistency mapping approaches are also implemented in Matlab, making those approaches more accessible to the community. The analysis is used to compare drug treatment (BFA and Lat b), abiotic stress (heat stress), and transgene expression (perhaps overexpression of a protein when the endogenous gene has not been knocked out) of ER shaping proteins on ER shape and movement. More sophisticated statistical tools (canonical variates analysis) are used to illustrate statistical differences that are actually fairly clear using other statistical approaches (e.g. Tukey two-way analysis), but these are handy and are included with the suite of software tools. These tools should be of value in other systems, such as cultured animal cells, where the ER network, or a substantial portion thereof, is relatively planar.

The main problem with this paper is that much of it has already been published in an uncited article by the Fricker group in the book edited by two of the co-authors, Kriechbaumer and Hawes: Fricker et al. (2018) Quantitation of ER Structure and Function. In *The Plant Endoplasmic Reticulum: Methods and Protocols*. (eds. C. Hawes and V. Kriechbaumer) pp. 43-66. This is quite an oversight and would have to be remedied by a fairly extensive re-write prior to publication. Many of the tools are described, and described more completely, in that paper: enhancing tubule elements, estimations of tubule size, automated segmentation and procedures for masking, and measuring bulges and constrictions in tubules. The supplemental figure is a slight modification of Figure 1 in that paper. Most of Figure 4 in that paper addresses some of the work in Figures 1 and 2 in this paper. As a matter of note in this paper, in Figure 2, the analysis of bulge and constrictions of ER should be compared between GFP-HDEL alone, and then RFP-RTN1 coexpressed with GFP-HDEL.

What isn't covered in the Fricker et al. (2018) is the application of these approaches to the drug, stress, and expression treatments above. Also this new paper has a re-implementation of persistency mapping in Matlab which very convincingly repeats of the Sparkes et al (2009) results for the drug treatments and now compares these with heat stress and Lunapark expression. The texture mapping and analysis of cisternal structure in Figure 3 is new, but is descriptive without mechanism (except for the Lunapark, and the mechanism there remains hypothetical in plants). Figure 4 and the movie are quite nice, particularly in comparing the persistency mapping with the optic flow. However, in this comparison it is apparent that rapidly streaming regions of tubules (according to the text), red and yellow in Figure 4j, are shown as persistent cisternae in Figure 4d. This could be overcome by subtracting the tubule signals from the cisternal signals prior to the persistency mapping for each image in the time series (an approach implemented in ref. 41, Figures 3 and 6). A real strength of this paper is the implementation of a new optical flow algorithm, but its strength lies in the local evaluation of flows in relatively persistent structures, an approach not taken here. Combining persistency and flow to clearly evaluate the flow within, rather than of, these structures would be quite enlightening. The average values presented in Figure 5b support the persistency values in 5a: in these treatments everything slows down. A very interesting result, which the authors mention, but do not dwell on, is that BFA does not increase the persistency of the tubules, a finding also made by Sparkes et al. (2009), even though cisternalization increases and the measured cisternae slow. The authors might hypothesize a mechanism for this, since one of the authors (Hawes) has been involved in debates of the cellular effects of BFA for more than twenty five years. As mentioned above, the statistical approaches shown in Figure 6 are welcome and will come in handy for such a pleomorphic and dynamic

organelle, even though most of the actual results (except for the Lunapark) can be inferred from other statistical approaches.

The microscopy is excellent, but there is a limitation on the work shown in Figure 2 which should be addressed. The thin optical sections that they use help to achieve the best x-y-z resolution, but in so-doing, might cause the z-dimensional out-of-the plane of section material to either be emphasized or minimized. This could artifactually give rise to the bulges and constrictions in the images of tubules. We know from EM that tubules do not have a constant distance from the plasma membrane/wall. It would be good to show, at least in some cases, that by increasing the size of the optical sections, or by over-sampling past the Nyquist criterion in the z, that the same bulges and constrictions can be visualized and are not a consequence of a cortical ER that varies in the z dimension from the cell surface.

Finally, the black-color intensity scales in the Figure 4f are not so good unless a light background is used. One can simply not see low-persistence structures in Figure 4f.

Reviewer #2 (Remarks to the Author):

The manuscript by Charlotte Pain and colleagues with title: "AnalyzeER: Quantitative analysis of plant ER architecture and dynamics" is a nicely done work. The authors developed a very important and innovative tool for the scientific community, especially for researchers investigating the endoplasmic reticulum morphology and dynamics. The accuracy of the software provided allows identification of multiple parameters and it can be applied to study effect of drugs, biotic, abiotic and other conditions on ER membrane morphology and dynamics.

Minor concern:

the authors should include more detailed Materials and Methods.

Reviewer #3 (Remarks to the Author):

The authors describe a novel software solution to extract a large number of quantitative measures describing both the structure and dynamics of the endoplasmic reticulum (ER) in plant cells. This is a very worthwhile endeavour since the complexity of the ER has previously limited experimenters to qualitative analyses or relatively simple, often manual measurements. The packaging of the entire measurement workflow into a single software package should make this type of analysis more accessible to a wide range of cell biologists. For the most part, the manuscript does an adequate job of explaining the complex image analysis that is being performed, but there are a number of places where I wish the text would go into more detail.

The quantification of proteins along the length of tubules (Figure 2) is interesting, but the results are presented in a confusing manner and are therefore not as clear as they could be. For example, the troughs in signal intensity for the RFP marker in Figure 2b should not be labeled as constrictions since they actually are thought to correspond to bulges in the tubes. Similarly, the legend for panel c should not describe the peaks of the respective markers as "bulges" since this implies that the width of the tubules varies for the different markers at different places. Furthermore, it is not clear why the ratios of signal intensities for GFP and RFP markers are not simply the inverse of each other in the table of Figure 2d. Finally, the bulge and constriction separations also do not seem to match the bulge and constriction densities (both in 2d). This last point may actually point to a problem that is hidden in the global averages listed in Figure 2d. Is it possible individual ER tubules do not conform to the general pattern described in the text, and that this pattern really only emerges by averaging over many tubules? This should be relatively easy to test by calculating the coincidence (within FWHM) of GFP peaks and RFP troughs, and vice versa.

I disagree with some of the comments made in the section of protein distribution within cisternae. The interpretation in line 229 that the low GFP-HDEL signal in cisternae expressing the RFP-LNP1

construct is caused by "appression of the two membrane faces" is not supported by any experimental evidence. Ideally, this should be tested with electron microscopy to obtain precise numbers, but as a first step, it should be possible to use a comparison of the signal intensities in normal and LNP1-overexpression cisternae. Similarly, the interpretation of the "small flecks of GFP-HDEL" within the cisternal sheets as "nano-scale holes" seems contrived. A much simpler explanation could be a local depletion of LNP1 which could lead to a local bulging of the ER membrane. This could be tested by measuring the intensity profiles of both GFP-HDEL and RFP-LNP1 across those small flecks, similar to Figure 3g.

The analysis of cisternal substructure (lines 233 ff) is an interesting approach to identify subtle effects on ER structure or protein distribution within the ER. However, simple listing of the four parameters (contrast, correlation, energy, homogeneity) will not provide meaningful insights to most cell biologists without additional explanations. For example, the fact that BFA treatment leads to an increase in "cisternal energy" is attributed to the presence of "BFA bodies" but it is not clear why these brighter structures should only affect this parameter and not, for example, homogeneity, which actually increases in these samples. How can the "energy" parameter be interpreted in a biological sense? (This also applies to the other three parameters.)

I do not understand what the authors mean by "BFA bodies" (line 251). In plant cells, BFA bodies accumulate post-Golgi cargo such as recycling plasma membrane proteins or the endocytosis tracer FM4-64. Using the same term for changes in ER organisation is confusing. It is also not clear why Figure 6b shows very bright accumulations of GFP-HDEL when Sparkes et al (2010) only described increased numbers and areas of cisternae. Is this an artefact of the very high BFA concentration used here?

I'm astonished that the authors cite almost exclusively papers from the Hawes lab or from Hawes lab alumni. Whilst this group has clearly made major contributions to our understanding of the ER, there are a number of other papers that provide insight into the organisation and dynamics of this organelle. I can easily think of the following papers: Ridge et al (1999) *Plant & Cell Physiology* 40: 1253-1261; Kang and Staehelin (2008) *Protoplasma* 234: 51-64; Jaipargas et al (2015) *Frontiers in Plant Sciences* 6:783; McFarlane et al (2017) *Plant & Cell Physiology* 58: 478-484.

Minor issues:

line 48: The reference for "nanoholes" (18) has not been published yet and should not be cited.

line 165: I'm astonished that the authors use zinc-iodide-osmium staining after aldehyde fixation to obtain measurements of ER tubule diameters and cisternal thickness. This procedure is well-known to induce artefacts such as swelling or membrane deformations. There are numerous publications available that use EM tomography after high-pressure freezing followed by freeze substitution that could provide more accurate measurements.

line 273: It is not clear how tubules or cisternae can move over fixed nodes. If the tubules and cisternae shift from time point to time point, then the nodes that connect them cannot remain fixed.

line 276: Supplementary movie 2 should be movie 1b.

line 277: black histograms should be white histograms

line 293: Supplementary movie 3 should be movie 1c.

Figure 1g: The structures labeled as ER tubules and cisternae cannot be recognised as such. Please provide images with higher magnification.

Figure 1l+m: The outlines of the cisternae do not match the apparent shapes of cisternae in panel k. The outlines appear to be shifted down and in some cases miss parts of the cisterna. For example, cisternae 3 and 5 in panel m appear to be connected in the micrograph of panel k in two places and enclose an open polygonal region.

Figure 2b: The distance should be given in micrometres, not in pixels.

Figure 2c: What do the white circles label?

AnalyzER: Quantitative analysis of plant ER architecture and dynamics.

Response to referees

Reviewer #1 (responses in italics)

This paper describes a Matlab-based, integrated set of software tools to analyze the changes in the [cortical] ER as it moves across the paradermal cytoplasm of mature plant epidermal cells and transitions between tubules to sheets. It incorporates interesting approaches to estimate the sub-resolution diameter of the ER and bases those estimations on measurements of serial block-face 3D reconstructions from ZIO impregnated tobacco leaf. Some improvements of the existing analysis techniques include a treatment of brightly-fluorescent ER bodies that move along with the ER tubules in Arabidopsis (but Arabidopsis is not included in the plant material section) and implementation of a better optical flow algorithm (Farnebäck) for looking at movement of and within tubules. Persistency mapping approaches are also implemented in Matlab, making those approaches more accessible to the community. The analysis is used to compare drug treatment (BFA and Lat b), abiotic stress (heat stress), and transgene expression (perhaps overexpression of a protein when the endogenous gene has not been knocked out) of ER shaping proteins on ER shape and movement. More sophisticated statistical tools (canonical variates analysis) are used to illustrate statistical differences that are actually fairly clear using other statistical approaches (e.g. Tukey two-way analysis), but these are handy and are included with the suite of software tools. These tools should be of value in other systems, such as cultured animal cells, where the ER network, or a substantial portion thereof, is relatively planar.

1. *We thank the referee for these positive comments. We have now included Arabidopsis in the Material and Methods.*

The main problem with this paper is that much of it has already been published in an uncited article by the Fricker group in the book edited by two of the co-authors, Kriechbaumer and Hawes: Fricker et al. (2018) Quantitation of ER Structure and Function. In *The Plant Endoplasmic Reticulum: Methods and Protocols*. (eds. C. Hawes and V. Kriechbaumer) pp. 43-66. This is quite an oversight and would have to be remedied by a fairly extensive re-write prior to publication. Many of the tools are described, and described more completely, in that paper: enhancing tubule elements, estimations of tubule size, automated segmentation and procedures for masking, and measuring bulges and constrictions in tubules. The supplemental figure is a slight modification of Figure 1 in that paper. Most of Figure 4 in that paper addresses some of the work in Figures 1 and 2 in this paper. As a matter of note in this paper, in Figure 2, the analysis of bulge and constrictions of ER should be compared between GFP-HDEL alone, and then RFP-RTN1 coexpressed with GFP-HDEL.

2. *The referee comments that much of the work has already been published, and suggests an extensive re-write is required to incorporate this material. We believe that the referee may not have appreciated that we provided very detailed information on all the existing and new methods in the form of a 114 page manual, and a tutorial to accompany the paper. This was made available on the Oxford Research Archive for download, and referenced in the paper, and we have assumed would form part of the evaluation along with download of the software itself. Whilst the methods in the paper submitted cover the analysis relevant to the figures presented, the manual is much more comprehensive and also provides the theoretical background to each processing step, such as the phase congruency analysis, and a description of all the*

options available for the entire processing pipeline. In a complementary manner, the tutorial provides a minimal set of instructions (and the data sets), for a user to apply the pipeline, using typical settings to illustrate the different analysis approaches. We agree with the referee that there is some overlap with Fig. 1 and 2 in this paper and the figures in the book chapter, but would argue that this is entirely appropriate given that all the functionality of the old program is included (and updated) in the AnalyzER program.

- 3. We now cite the book chapter and document the improvements made in the AnalyzER program at the end of the introduction, and provide both the manual and tutorial as part of the supplementary material. Supplementary figure 2 has also been completely redrawn to highlight the additional measurements of morphology and dynamics in the AnalyzER program.*
- 4. We have included analysis of GFP-HDEL alone along with a statistical comparison of GFP-HDEL in the presence of RFP-RTN1, as suggested by the referee.*

What isn't covered in the Fricker et al. (2018) is the application of these approaches to the drug, stress, and expression treatments above. Also this new paper has a re-implementation of persistency mapping in Matlab which very convincingly repeats of the Sparkes et al (2009) results for the drug treatments and now compares these with heat stress and Lunapark expression. The texture mapping and analysis of cisternal structure in Figure 3 is new, but is descriptive without mechanism (except for the Lunapark, and the mechanism there remains hypothetical in plants). Figure 4 and the movie are quite nice, particularly in comparing the persistency mapping with the optic flow. However, in this comparison it is apparent that rapidly streaming regions of tubules (according to the text), red and yellow in Figure 4j, are shown as persistent cisternae in Figure 4d. This could be overcome by subtracting the tubule signals from the cisternal signals prior to the persistency mapping for each image in the time series (an approach implemented in ref. 41, Figures 3 and 6).

- 5. We note that the tubule signals are already subtracted from the cisternal signals as the referee has suggested, and speed and persistency of both are quantified independently. The reason why some parts of the rapidly streaming regions are classified as cisternal structures, is because any tubular elements are not sufficiently well resolved. We comment on this general problem in the introduction, and also note that it is easy to partition cisternae into different classes based on their morphology and dynamics (or indeed texture). We include an additional Supplementary figure 6, which illustrates this approach.*

A real strength of this paper is the implementation of a new optical flow algorithm, but its strength lies in the local evaluation of flows in relatively persistent structures, an approach not taken here. Combining persistency and flow to clearly evaluate the flow within, rather than of, these structures would be quite enlightening.

- 6. We thank the referee for this comment and we have now updated the analysis to include local measurements of the flow coherence (Fig. 5c), as well as curl and divergence of the vector field along tubules and within the cisternae (Supplementary Figs 4&5). The methods in the paper and manual are also updated. The flow coherence for tubules and cisternae has now been included in a revised set of statistics for the MANOVA (Fig. 6), and shows highly significant differences between the control and treatments.*

The average values presented in Figure 5b support the persistency values in 5a: in these treatments everything slows down. A very interesting result, which the authors mention, but do not dwell on, is that BFA does not increase the persistency of the tubules, a finding also made by Sparkes et al. (2009), even though cisternalization increases and the measured cisternae slow. The authors might hypothesize a mechanism for this, since one of the authors (Hawes) has been involved in debates of the cellular effects of BFA for more than twenty five years. As mentioned above, the statistical approaches shown in Figure 6 are welcome and will come in handy for such a pleomorphic and dynamic organelle, even though most of the actual results (except for the Lunapark) can be inferred from other statistical approaches.

7. *We agree with the referee that there are many new observations that arise from the comprehensive analysis. However, the purpose of this paper is to describe the methods and software. Biological interpretation of this, and many other data sets, is the subject of future publications.*

The microscopy is excellent, but there is a limitation on the work shown in Figure 2 which should be addressed. The thin optical sections that they use help to achieve the best x-y-z resolution, but in so-doing, might cause the z-dimensional out-of-the plane of section material to either be emphasized or minimized. This could artifactually give rise to the bulges and constrictions in the images of tubules. We know from EM that tubules do not have a constant distance from the plasma membrane/wall. It would be good to show, at least in some cases, that by increasing the size of the optical sections, or by over-sampling past the Nyquist criterion in the z, that the same bulges and constrictions can be visualized and are not a consequence of a cortical ER that varies in the z dimension from the cell surface.

8. *The comment that the referee makes on the risk of signal fluctuations due to the ER tubules moving out-of-the plane of focus in Fig. 2 would have been an entirely justified criticism of the original, single channel program, but is dealt with far more effectively here through dual-labelling. Thus the data reported in Fig. 2 show the diminution of the luminal HDEL signal (green) is tightly correlated with the presence of the tagged-reticulon (magenta), which also demonstrates clearly that the tubule is still being imaged correctly and is in the focal plane. We would argue that using dual-labelling and maintaining super-resolution as shown in Fig. 2 is a more robust solution than either opening the pinhole to lower resolution, or collecting z-series from such highly dynamic structures that would yield movement artefacts (unless the actin cytoskeleton is pharmacologically depolymerised).*

Finally, the black-color intensity scales in the Figure 4f are not so good unless a light background is used. One can simply not see low-persistency structures in Figure 4f.

9. *We thank the referee for this comment and we have redone the figures with a grey background as suggested, which makes regions of persistency much clearer. We have also updated the software and manual to reflect the additional checkbox option in the GUI to set the background to grey.*

Reviewer #2 (Remarks to the Author):

The manuscript by Charlotte Pain and colleagues with title: “AnalyzeER: Quantitative analysis of plant ER architecture and dynamics” is a nicely done work. The authors developed a very important and innovative tool for the scientific community, especially for researchers investigating the endoplasmic reticulum morphology and dynamics. The accuracy of the software provided allows identification of multiple parameters and it can be applied to study

effect of drugs, biotic, abiotic and other conditions on ER membrane morphology and dynamics.

Minor concern:

the authors should include more detailed Materials and Methods.

10. We thank the referee for this positive review. Very detailed methods are now included in the manual that now forms part of the Supplementary material.

Reviewer #3 (Remarks to the Author):

The authors describe a novel software solution to extract a large number of quantitative measures describing both the structure and dynamics of the endoplasmic reticulum (ER) in plant cells. This is a very worthwhile endeavour since the complexity of the ER has previously limited experimenters to qualitative analyses or relatively simple, often manual measurements. The packaging of the entire measurement workflow into a single software package should make this type of analysis more accessible to a wide range of cell biologists. For the most part, the manuscript does an adequate job of explaining the complex image analysis that is being performed, but there are a number of places where I wish the text would go into more detail.

11. We thank the referee for this positive review. Again, very detailed methods are included in the manual that now forms part of the Supplementary material.

The quantification of proteins along the length of tubules (Figure 2) is interesting, but the results are presented in a confusing manner and are therefore not as clear as they could be. For example, the troughs in signal intensity for the RFP marker in Figure 2b should not be labeled as constrictions since they actually are thought to correspond to bulges in the tubes. Similarly, the legend for panel c should not describe the peaks of the respective markers as "bulges" since this implies that the width of the tubules varies for the different markers at different places. Furthermore, it is not clear why the ratios of signal intensities for GFP and RFP markers are not simply the inverse of each other in the table of Figure 2d. Finally, the bulge and constriction separations also do not seem to match the bulge and constriction densities (both in 2d). This last point may actually point to a problem that is hidden in the global averages listed in Figure 2d. Is it possible individual ER tubules do not conform to the general pattern described in the text, and that this pattern really only emerges by averaging over many tubules? This should be relatively easy to test by calculating the coincidence (within FWHM) of GFP peaks and RFP troughs, and vice versa.

12. The referee is correct to point out this ambiguity. We now refer to peaks and troughs in the intensity traces and only refer to bulges and constrictions for the GFP-HDEL channel. We have changed the layout of the figure and also included the comparison with the GFP-HDEL alone requested by referee 1.

13. The values for densities and separations are unlikely to agree as peaks/trough density is measured along the total length of the tubule, whilst separation is only measured where there are at least two peaks/troughs along the tubule (i.e. excluding the segment of tubule linking to a node).

14. The ratios of GFP/RFP and RFP/GFP are not simple inverses as they are measured at the corresponding peak for each channel.

I disagree with some of the comments made in the section of protein distribution within cisternae. The interpretation in line 229 that the low GFP-HDEL signal in cisternae

expressing the RFP-LNP1 construct is caused by "appression of the two membrane faces" is not supported by any experimental evidence. Ideally, this should be tested with electron microscopy to obtain precise numbers, but as a first step, it should be possible to use a comparison of the signal intensities in normal and LNP1-overexpression cisternae. Similarly, the interpretation of the "small flecks of GFP-HDEL" within the cisternal sheets as "nano-scale holes" seems contrived. A much simpler explanation could be a local depletion of LNP1 which could lead to a local bulging of the ER membrane. This could be tested by measuring the intensity profiles of both GFP-HDEL and RFP-LNP1 across those small flecks, similar to Figure 3g.

15. We have deleted the comment as to the potential mode of action of RFP-LNP1 as we do not have the corresponding EM evidence.

16. We have included the suggested from the referee that bulging may arise from local depletion of LNP1

The analysis of cisternal substructure (lines 233 ff) is an interesting approach to identify subtle effects on ER structure or protein distribution within the ER. However, simple listing of the four parameters (contrast, correlation, energy, homogeneity) will not provide meaningful insights to most cell biologists without additional explanations. For example, the fact that BFA treatment leads to an increase in "cisternal energy" is attributed to the presence of "BFA bodies" but it is not clear why these brighter structures should only affect this parameter and not, for example, homogeneity, which actually increases in these samples. How can the "energy" parameter be interpreted in a biological sense? (This also applies to the other three parameters.)

17. We have included more explanation of the texture feature terms in the text and methods to help clarify this issue. Equally, these are standard metrics used to characterise texture that are widely used. We have included more explanation and references to the texture metrics, including ¹, recommendations for meaningful comparisons using GLCM ², and provide more context with other applications in biology ³.

I do not understand what the authors mean by "BFA bodies" (line 251). In plant cells, BFA bodies accumulate post-Golgi cargo such as recycling plasma membrane proteins or the endocytosis tracer FM4-64. Using the same term for changes in ER organisation is confusing.

18. We agree with the referee and have deleted references to BFA bodies.

It is also not clear why Figure 6b shows very bright accumulations of GFP-HDEL when Sparkes et al (2010) only described increased numbers and areas of cisternae. Is this an artefact of the very high BFA concentration used here?

19. We do not think the concentration of BFA used here (100 µg ml⁻¹ for 1h) gives artefactual results – Whilst Sparkes et al. (2010) used 50 µg ml⁻¹ for 30 min, previously Sparkes et al. (2009) used 100 µg ml⁻¹ for 3 h. We think that the variation in intensity reflects improved imaging with the AiryScan system and avoidance of saturation to allow quantitative imaging.

I'm astonished that the authors cite almost exclusively papers from the Hawes lab or from Hawes lab alumni. Whilst this group has clearly made major contributions to our understanding of the ER, there are a number of other papers that provide insight into the

organisation and dynamics of this organelle. I can easily think of the following papers: Ridge et al (1999) *Plant & Cell Physiology* 40: 1253-1261; Kang and Staehelin (2008) *Protoplasma* 234: 51-64; Jaipargas et al (2015) *Frontiers in Plant Sciences* 6:783; McFarlane et al (2017) *Plant & Cell Physiology* 58: 478-484.

20. We have included more references as requested by the referee including reviews of early work on ER⁴⁻⁶, additional information on movement⁷, membrane contact sites⁸ and ER bodies^{9,10}

Minor issues:

line 48: The reference for "nanoholes" (18) has not been published yet and should not be cited.

21. We have deleted this reference

line 165: I'm astonished that the authors use zinc-iodide-osmium staining after aldehyde fixation to obtain measurements of ER tubule diameters and cisternal thickness. This procedure is well-known to induce artefacts such as swelling or membrane deformations. There are numerous publications available that use EM tomography after high-pressure freezing followed by freeze substitution that could provide more accurate measurements.

*22. We disagree with the referee on this issue. ZIO staining is preceded by conventional fixation which stabilises the membranes and there are no artefacts induced by the subsequent ZIO treatment. This is a very old argument and is simply untrue as evidenced by the data presented. Indeed, extended osmium impregnation techniques after aldehyde fixation are the standard adopted by the SBF-SEM community.. Conversely, high pressure freezing does indeed cause the plant ER to dilate and material has too little contrast for SBF-SEM use so it simply cannot be used for this kind of work or for making any meaningful measurements. What we have shown is that conventional fixation followed by osmium enhancement of membranes is in fact an excellent technique. We reiterate that the entire SBF-SEM community use extended osmium impregnation techniques after aldehyde fixation if there is no pre-embedding selective contrasting eg. Puhka et al. *Mol. Biol. Cell* 2012, 23, 242. Holcombe et al (2012) *J Neuroscience* 33, 12945.*

line 273: It is not clear how tubules or cisternae can move over fixed nodes. If the tubules and cisternae shift from time point to time point, then the nodes that connect them cannot remain fixed.

23. This is a common observation of ER dynamics and is interpreted as membrane flow

line 276: Supplementary movie 2 should be movie 1b.

24. We have corrected all the references to the supplementary movies in the text

line 277: black histograms should be white histograms

25. We have corrected the labelling

line 293: Supplementary movie 3 should be movie 1c.

26. We have corrected all the references to the supplementary movies in the text

Figure 1g: The structures labeled as ER tubules and cisternae cannot be recognised as such. Please provide images with higher magnification.

27. We have reorganised the figure to show how the tubules and cisternae are identified from the 3D reconstruction and also provided higher magnification images to show them in cross-section.

Figure 11+m: The outlines of the cisternae do not match the apparent shapes of cisternae in panel k. The outlines appear to be shifted down and in some cases miss parts of the cisterna. For example, cisternae 3 and 5 in panel m appear to be connected in the micrograph of panel k in two places and enclose an open polygonal region.

28. We have corrected the mis-alignment that was introduced during figure composition.

Figure 2b: The distance should be given in micrometres, not in pixels.

29. We have changed the axis scaling as suggested by the referee

Figure 2c: What do the white circles label?

30. We have re-formatted the figure to make it clearer and updated the legend. The white circles have now been replaced with cyan circles and mark the nodes.

Additional references included in the main text:

- 1 Haralick, R. M. Statistical and structural approaches to texture. *Proceedings of the IEEE* **67**, 786-804, doi:10.1109/PROC.1979.11328 (1979).
- 2 Brynolfsson, P. *et al.* Haralick texture features from apparent diffusion coefficient (ADC) MRI images depend on imaging and pre-processing parameters. **7**, 4041 (2017).
- 3 Boland, M. V. & Murphy, R. F. A neural network classifier capable of recognizing the patterns of all major subcellular structures in fluorescence microscope images of HeLa cells. *Bioinformatics* **17**, 1213-1223, doi:DOI 10.1093/bioinformatics/17.12.1213 (2001).
- 4 Hepler, P. K., Palevitz, B. A., Lancelle, S. A., McCauley, M. M. & Lichtscheidl, I. K. Cortical endoplasmic reticulum in plants. *J. Cell Sci.* **96**, 355-373 (1990).
- 5 Staehelin, L. A. The plant ER: a dynamic organelle composed of a large number of discrete functional domains. *Plant J.* **11**, 1151-1165, doi:10.1046/j.1365-313X.1997.11061151.x (1997).
- 6 Ridge, R. W., Uozumi, Y., Plazinski, J., Hurley, U. A. & Williamson, R. E. Developmental transitions and dynamics of the cortical ER of Arabidopsis cells seen with green fluorescent protein. *Plant Cell Physiol.* **40**, 1253-1261 (1999).
- 7 Lichtscheidl, I. K. & Url, W. G. Organization and dynamics of cortical endoplasmic reticulum in inner epidermal cells of onion bulb scales. *Protoplasma* **157**, 203-215, doi:10.1007/BF01322653 (1990).
- 8 McFarlane, H. E. *et al.* Multiscale Structural Analysis of Plant ER-PM Contact Sites. *Plant Cell Physiol.* **58**, 478-484, doi:10.1093/pcp/pcw224 (2017).

- 9 Matsushima, R. *et al.* The ER body, a novel endoplasmic reticulum-derived structure in *Arabidopsis*. *Plant Cell Physiol.* **44**, 661-666 (2003).
- 10 Nakano, R. T., Yamada, K., Bednarek, P., Nishimura, M. & Hara-Nishimura, I. J. F. i. p. s. ER bodies in plants of the Brassicales order: biogenesis and association with innate immunity. **5**, 73 (2014).

REVIEWERS' COMMENTS:

Reviewer #1 (Remarks to the Author):

The authors have completely addressed my concerns.
Lawrence Griffing

Reviewer #3 (Remarks to the Author):

I thank the authors for the revisions of the text and figures. The additional explanations will make the paper (and the software) more accessible to cell biologists. I have no further comments, aside from one minor point for correction: Reference 73 (Brynolfsson) is missing the journal name.

Of note, I have neither reviewed the 114 page manual that was now provided in the supplement nor have I tested the software which is apparently available on the authors' website. Full evaluation of the software for its usefulness and accuracy would be desirable but clearly goes beyond the scope of this manuscript review.

REVIEWERS' COMMENTS:

Reviewer #1 (Remarks to the Author):

The authors have completely addressed my concerns.
Lawrence Griffing

Reviewer #3 (Remarks to the Author):

I thank the authors for the revisions of the text and figures. The additional explanations will make the paper (and the software) more accessible to cell biologists. I have no further comments, aside from one minor point for correction: Reference 73 (Brynolfsson) is missing the journal name.

- We have added the journal name as requested